# RORα controls hepatic lipid homeostasis via negative regulation of PPARγ transcriptional network

Kyeongkyu Kim[1], Kyungjin Boo[1], Young Suk Yu[1], Se Kyu Oh[1], Hyunkyung Kim[1], Yoon Jeon[2], Jinhyuk Bhin[3], Daehee Hwang[3], Keun Il Kim [4], Jun-Su Lee[5], Seung-Soon Im[5], Seul Gi Yoon[6,7], Il Yong Kim[6,7], Je Kyung Seong[6,7,8], Ho Lee[2], Sungsoon Fang[9] & Sung Hee Baek[1]

The retinoic acid receptor-related orphan receptor-α (RORα) is an important regulator of various biological processes, including cerebellum development, circadian rhythm and cancer. Here, we show that hepatic RORα controls lipid homeostasis by negatively regulating transcriptional activity of peroxisome proliferators-activated receptor-γ (PPARγ) that mediates hepatic lipid metabolism. Liver-specific *Rorα*-deficient mice develop hepatic steatosis, obesity and insulin resistance when challenged with a high-fat diet (HFD). Global transcriptome analysis reveals that liver-specific deletion of *Rorα* leads to the dysregulation of PPARγ signaling and increases hepatic glucose and lipid metabolism. RORα specifically binds and recruits histone deacetylase 3 (HDAC3) to PPARγ target promoters for the transcriptional repression of PPARγ. PPARγ antagonism restores metabolic homeostasis in HFD-fed liver-specific *Rorα* deficient mice. Our data indicate that RORα has a pivotal role in the regulation of hepatic lipid homeostasis. Therapeutic strategies designed to modulate RORα activity may be beneficial for the treatment of metabolic disorders.

[1] Department of Biological Sciences, Creative Research Initiatives Center for Chromatin Dynamics, Seoul National University, Seoul 08826, South Korea. [2] Graduate School of Cancer Science and Policy, Research Institute, National Cancer Center, Gyeonggi-do 10408, South Korea. [3] Department of New Biology and Center for Plant Aging Research, Institute for Basic Science, DGIST, Daegu 42988, South Korea. [4] Department of Biological Sciences, Sookmyung Women's University, Seoul 04310, South Korea. [5] Department of Physiology, Keimyung University School of Medicine, Daegu 42601, South Korea. [6] Laboratory of Developmental Biology and Genomics, College of Veterinary Medicine, Research Institute for Veterinary Science, Seoul National University, Seoul 08826, South Korea. [7] Korea Mouse Phenotyping Center, Seoul 08826, South Korea. [8] BK21 Plus Program for Creative Veterinary Science Research, BIO-MAX institute, Interdisciplinary Program for Bioinformatics and Program for Cancer Biology, Seoul National University, Seoul 08826, South Korea. [9] Severance Biomedical Science Institute, BK21 Plus Project for Medical Science, Gangnam Severance Hospital, Yonsei University College of Medicine, Seoul 06273, South Korea. Kyeongkyu Kim and Kyungjin Boo contributed equally to this work. Correspondence and requests for materials should be addressed to S.F. (email: sfang@yuhs.ac) or to S.H.B. (email: sbaek@snu.ac.kr)

O besity is a high-risk metabolic disorder, leading to various complications, including cardiovascular disease, hyperlipidemia and type II diabetes[1–3]. Ectopic accumulation of fat in various tissues activates numerous cellular stress and inflammatory signaling pathways, resulting in insulin resistance, pancreatic β-cell dysfunction, and hepatic steatosis[4]. The liver is the central metabolic organ to regulate key aspects of glucose and lipid metabolism including gluconeogenesis, fatty acid β-oxidation, lipoprotein uptake and secretion and lipogenesis[5]. Given that portal vein is a critical path to convey insulin signaling from pancreas during fed state, the hepatic glucose and lipid metabolism are directly under control of nutrient signaling.

Dysregulation of hepatic lipid metabolism results in the development of hepatic steatosis, contributing to the chronic insulin resistance and steatotic hepatitis[6, 7]. The hepatic metabolic pathways are governed by highly dynamic transcriptional networks of orphan nuclear receptors (ONRs), including proliferators-activated receptor-γ (PPARγ), farnesoid X receptor, and liver X receptor[8]. ONRs are ligand-activated transcription factors with no defined ligands[9, 10]. Many ONRs are expressed in tissues involved in metabolism, such as skeletal muscle, adipose tissue and liver[11, 12], and play critical roles in the regulation of metabolism[13]. Genetic studies have shown that many ONRs regulate nutrient metabolism and physiology of obesity and type II diabetes[14–16]. Given that numerous synthesized ligands for ONRs are used for developing putative drugs for human metabolic diseases[17–19], ONRs are emerging as therapeutic targets for the treatment of metabolic diseases.

Previously, we have reported that receptor-related orphan receptor-α (RORα), a member of ONRs, possesses tumor suppressive function by transrepressing canonical Wnt/β-catenin signaling leading to inhibition of colon cancer growth and by increasing p53 stability upon DNA damage response[20, 21]. RORα is known to regulate cerebellum development[22]. The *staggerer* (*sg*) mice, natural *Rorα* spontaneous mutant mice, display ataxia and severe cerebellar atrophy[23]. Moreover, RORα functions to regulate circadian rhythm as a key regulator of the cyclic expression of BMAL1 together with REV-ERBα[24]. The RORα/REV-ERBα feedback loop controls the circadian expression pattern of BMAL1, indicating that RORα plays a key role in the core circadian clock[25]. In addition, *sg* mice show lower expression levels of genes involved in lipid metabolism, including apolipoprotein A-1 (*apoA1*) and apolipoprotein C-III (*apoCIII*)[26, 27]. Thus, *sg* mice exhibit less body weight gain compared with wild-type (WT) mice[28]. Given that *sg* mice have huge cerebellar defects, it is still possible that physiological changes observed in *sg* mice are indirect effects. Thus, the physiological roles of RORα to control transcriptional networks to modulate lipogenesis and gluconeogenesis still remain unclear.

Here, we report that RORα plays a key role to control hepatic lipid metabolism to protect against diet-induced obesity and hepatic steatosis, using liver-specific *Rorα*-deficient mouse model. High-fat diet (HFD)-fed liver-specific *Rorα* deficient mice (RORα^LKO mice) show severe metabolic defects, including hepatic steatosis, obesity, and insulin resistance, although no physiological changes have been observed with control diet (CD). Genome-wide transcriptome analysis reveals that PPARγ signaling is remarkably elevated in RORα^LKO mice. RORα specifically recruits HDAC3 to the PPARγ target promoters to suppress PPARγ transcriptional activity. Finally, PPARγ antagonism by using PPARγ antagonist GW9662, largely ameliorates body weight gain and hepatic steatosis in HFD-fed RORα^LKO mice, indicating that dysregulated PPARγ signaling is a critical metabolic cue, leading to metabolic defects in HFD-fed RORα^LKO mice. Together, our data demonstrate that RORα controls PPARγ signaling to protect against hepatic metabolic homeostasis and obesity in response to HFD.

## Results

### HFD induces obesity in liver-specific *Rorα*-deficient mice.
To determine the physiological roles of RORα in the liver, we generated RORα-floxed mice (RORα^f/f) by gene targeting in ES cells and crossed RORα floxed mice with Albumin-Cre (Alb-Cre) mice to selectively create liver-specific *Rorα* conditional knockout (KO) mice (hereafter named RORα^LKO) (Fig. 1a, b). The mRNA and protein levels of endogenous hepatic RORα were remarkably depleted in RORα^LKO mice compared with littermate controls (hereafter named RORα^f/f) (Fig. 1c–f). We measured the growth rate of RORα^LKO mice and observed that they attained body weights similar to RORα^f/f mice fed CD during 10 weeks from 8 weeks old (Fig. 1g). Body composition analysis revealed that RORα^LKO mice exhibited similar fat/lean mass, free body fluid and adipocytes size with those of RORα^f/f (Supplementary Fig. 1a, b). However, when placed on a HFD, RORα^LKO mice exhibited a significant increase of the weight gain (20 vs. 25 g) compared with their RORα^f/f littermates, resulting in extraordinary obesity (Fig. 1g). Body composition analysis and macroscopic view revealed that RORα^LKO mice had more fat mass (Fig. 1h, i). All white and brown fat depots from RORα^LKO mice were significantly increased in mass relative to RORα^f/f (Fig. 1j). During obesity, adipose tissue expands by hyperplastic and/or hypertrophic growth. The cross-sectional area of adipocytes in visceral fat tissue was markedly increased in RORα^LKO mice compared with RORα^f/f mice (Fig. 1k). Induction of pro-inflammatory genes, including *Mcp1*, *Ifnγ*, and *F4/80* in visceral fat depot were potentiated in RORα^LKO mice (Fig. 1l). Consistent with a significant weight gain in HFD-fed RORα^LKO mice, gene expression analysis revealed reduction of *Pgc1α*, as well as a number of genes involved in thermogenesis, mitochondrial biogenesis, and fatty acid oxidation in brown adipose tissue of RORα^LKO mice compared with that of RORα^f/f littermates (Fig. 1m). The observation that energy expenditure (EE) in brown fat has been impaired in HFD-fed RORα^LKO mice led us to examine whether they have global metabolic defects. Although no obvious defects were observed in mice on CD (Supplementary Fig. 1c), HFD-fed RORα^LKO mice were found to produce far less $CO_2$, consume less $O_2$ and expend less energy than RORα^f/f littermates, indicating that oxidative phosphorylation is impaired by the hepatic deletion of RORα (Fig. 1n and Supplementary Fig. 1d). Previously, bile acids have been reported to increase EE by promoting intracellular thyroid hormone activation in brown adipose tissue[29]. We observed that expression of key genes involved in hepatic bile acid synthesis was remarkably reduced in HFD-fed RORα^LKO mice (Supplementary Fig. 1e). Consistently, serum bile acid pool sizes in HFD-fed RORα^LKO mice were markedly less than RORα^f/f littermates (Supplementary Fig. 1f), implicating that reduction of bile acid synthesis and bile acid pool size led to reduced EE in brown adipose tissue of HFD-fed RORα^LKO mice.

### Hepatic steatosis impairs insulin sensitivity in RORα^LKO mice.
Obesity is largely associated with hepatic steatosis in humans as well as in rodents. Consistent with obese phenotype in RORα^LKO mice, large lipid vesicles with increased amounts were observed in the hepatocytes of HFD-fed RORα^LKO mice (Fig. 2a). Macroscopically, liver from HFD-fed RORα^LKO mice was markedly enlarged and paler compared with HFD-fed RORα^f/f liver (Fig. 2b). Consistently, HFD-fed RORα^LKO mice exhibited a remarkable increase of liver weight compared with HFD-fed RORα^f/f mice (Fig. 2c). In accordance with hematoxylin and eosin

staining, oil red O staining and hepatic triglyceride (TG) analysis showed a dramatic increase in lipid level in the HFD-fed RORα[LKO] liver compared with the HFD-fed RORα[f/f] liver, whereas no difference was observed in CD-fed RORα[f/f] and RORα[LKO] mice (Fig. 2d, e and Supplementary Fig. 1g). While

hepatic gene expression profiles were similar among CD-fed genotypes (Supplementary Fig. 1h), hepatic gene expression profiles of lipogenesis, gluconeogenesis, and lipid sequestration in the HFD-fed RORα[LKO] were largely increased, indicating that RORα protects against HFD-induced hepatic steatosis (Fig. 2f).

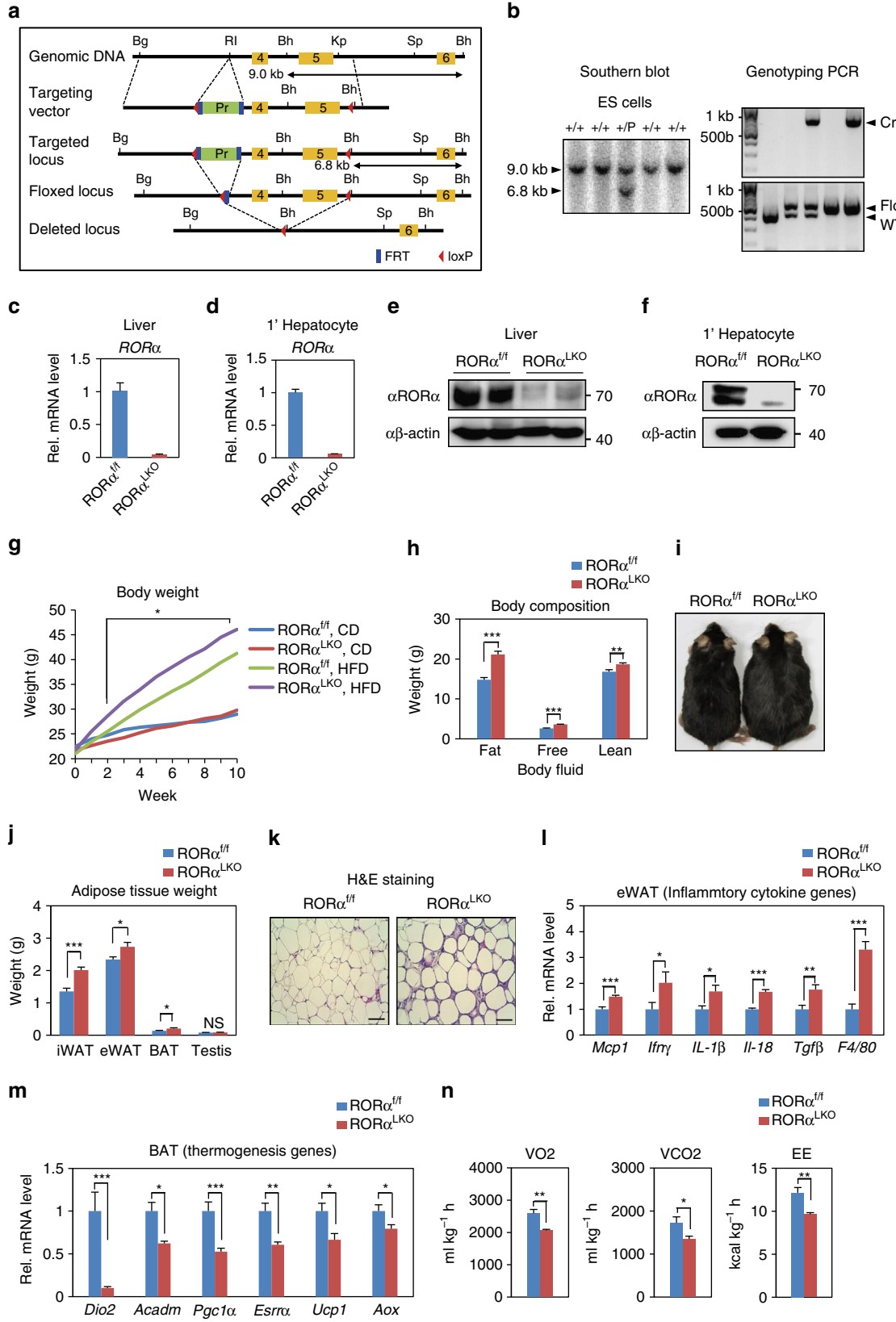

Obesity and hepatic steatosis often predispose rodents and humans to impaired glucose homeostasis and insulin resistance[30–32]. Hepatic deficiency of RORα resulted in elevated fasting insulin levels in RORα[LKO] mice (Fig. 2g). As elevated fasting insulin level is an indication of insulin resistance, RORα[LKO] mice predisposed to severe insulin resistance than RORα[f/f] mice. Consistent with elevated fasting insulin level, an investigation of insulin signaling pathways confirmed reduction of phosphorylated AKT, indicating that insulin signaling was impaired in HFD-fed RORα[LKO] mice (Fig. 2h). As insulin signaling was impaired in the liver, we performed glucose tolerance tests (GTTs) and insulin tolerance tests (ITTs) to determine if glucose homeostasis was impaired in HFD-fed RORα[LKO] mice. Glucose intolerance and insulin resistance were observed in HFD-fed RORα[LKO] mice, although CD-fed RORα[LKO] mice exhibited little or no difference in glucose homeostasis compared with CD-fed RORα[f/f] mice (Fig. 2i, j). Altogether, our data strongly demonstrate that hepatic RORα is required for prevention against insulin resistance.

**RORα[LKO] mice exhibit enhanced PPARγ transcriptional activity.** To explore molecular mechanism by which hepatic deletion of RORα induces obesity and insulin resistance, we performed mRNA-sequencing analysis of liver tissues obtained from HFD-fed RORα[f/f], HFD-fed RORα[LKO], CD-fed RORα[f/f] and CD-fed RORα[LKO] mice (Supplementary Data Table 1). Using the resulting mRNA expression profiles, we first identified 343 differentially expressed genes (DEGs; see Methods) between HFD-fed RORα[LKO] and HFD-fed RORα[f/f] mice (RORα[LKO]/RORα[f/f]$_{HFD}$ in Fig. 3a) and also 395 DEGs between CD-fed RORα[LKO] and CD-fed RORα[f/f] mice (RORα[LKO]/RORα[f/f]$_{CD}$ in Fig. 3a and Supplementary Data 2). Moreover, we further compared log$_2$-fold changes of the DEGs in the two comparisons above ((RORα[LKO]/RORα[f/f]$_{HFD}$)/(RORα[LKO]/RORα[f/f]$_{CD}$) in Fig. 3a) and identified the genes specifically affected by RORα under HFD condition as the DEGs showing significant ($P < 0.05$) differences in the log$_2$-fold changes (Supplementary Data 2). We categorized these DEGs into eight groups (Groups 1–8) based on differential expression patterns in the three comparisons above. Our data above showed that we only found significant weight gain of RORα[LKO] mice under HFD condition. Of Groups 1–8, thus, we first focused on Groups 1–4 showing significant changes under HFD condition (Fig. 3a).

To understand cellular processes represented by Groups 1–4, we performed enrichment analysis of gene ontology biological processes (GOBPs) and Kyoto Encyclopedia of Genes and Genomes (KEGG) pathways for the genes in Groups 1–4 using DAVID software [33, 34] (Supplementary Data 3). Group 1 is

mainly involved in the processes related to PPAR and adipocytokine signaling pathways and fatty acid/retinol metabolism (Fig. 3b). Group 4 is mainly involved in the processes related to circadian rhythm (Supplementary Data 3). Since Group 1 is highly associated with the weight gain of HFD-fed RORα[LKO], we next examined which transcription factors (TFs) account for up-regulation of the genes in Group 1 under HFD condition. By performing TF enrichment analysis of the genes in Group 1 using ChEA2 software[35], PPARγ turned out to be the most enriched TF in Group 1 (Fig. 3c and Supplementary Data 4). Quantitative PCR with reverese transcription analysis confirmed that the genes in Group 1 including PPARγ target genes are largely elevated in HFD-fed RORα[LKO] mice (Fig. 3d), indicating that PPARγ transcriptional activity is enhanced in the absence of RORα.

PPARα is a transcriptional factor that conducts a key role in hepatic lipid metabolism and shares similar response elements with PPARγ on the target promoters[36, 37]. To determine whether RORα also mediates PPARα transcriptional network in the liver, we examined the expression of well-known hepatic PPARα target genes, including *Acox1* and *Fgf21*. The hepatic gene expressions of *Acox1* and *Fgf21* in HFD-fed RORα[LKO] mice were similar to those of HFD-fed RORα[f/f] mice, suggesting that RORα deficiency would not further enhance hepatic PPARα transcriptional network (Supplementary Fig. 2a) under HFD condition. We next examined the expression of PPARα target genes in the physiological setting of PPARα activation. It has been widely accepted that PPARα is activated under conditions of energy deprivation[38]. The induction of PPARα target genes in RORα[LKO] mice was quite similar to that of RORα[f/f] mice (Supplementary Fig. 2b). Chromatin immunoprecipitation (ChIP) assay clearly revealed little or no difference of PPARα recruitment to PPAR-response element (PPRE) on the promoters of PPARα target genes (Supplementary Fig. 2c). Recently, PPARα has been reported to bind autophagic gene promoters to coordinate autophagy in response to nutrient deprivation[37]. We observed that the induction of autophagic genes including *LC3a* and *Sesn2* of RORα[LKO] mice were similar to those of RORα[f/f] mice (Supplementary Fig. 2d). Taken together, these data indicate that RORα mainly controls PPARγ transcriptional network rather than PPARα in the liver in response to environmental stress such as HFD.

**RORα represses PPARγ transcriptional activity via HDAC3.** Since PGC1α is a well-known coactivator for PPARγ[39], we examined whether introduction of RORα inhibits PPARγ/PGC1α-dependent transcriptional activation using PPRE-containing luciferase reporter. Expression of PGC1α

**Fig. 1** Liver-specific *Rorα* deleted mice are susceptible to diet-induced obesity. **a** Schematic representation of the *Rorα* gene-targeting strategy, including a map of the RORα exon 4 and 5 allele (*yellow box*) and the targeting vector with loxP sites (*red arrowhead*), FRT sites (*blue box*), and puromycin selection gene (*green box*). Bg: BglII, RI: EcoRI, Bh: BamHI, Kp: KpnI, Sp: SpeI. **b** Southern blot analysis to screen correctly targeted *Rorα* + /puro ES cell clones. For BamHI digestion, the bands representing WT and mutant alleles were 9.0 kb and 6.8 kb, respectively. PCR analyses with genomic DNA extracted from tail of WT, RORα[f/+], Alb; RORα[f/+], RORα[f/f] and Alb; RORα[f/f] mice are shown. PCR were performed to amplify the cre (*top*), floxed and deleted allele (*bottom*). **c**, **d** mRNA expression level of *RORα* in liver extract **c** and primary hepatocyte **d** from RORα[f/f] and RORα[LKO] mice. Expression was normalized to 18 s rRNA expression. **e**, **f** Protein expression level of RORα in liver extract **e** and primary hepatocyte **f**. **g** Body weight change in RORα[f/f] and RORα[LKO] mice fed CD or HFD for 10 weeks ($n = 9–12$/group). Statistical analysis was performed using Student's unpaired t-test. *$P < 0.05$, RORα[f/f] vs. RORα[LKO], HFD. **h**, **i** RORα[f/f] and RORα[LKO] mice were fed with HFD for 10 weeks. **h** Body composition analysis of RORα[f/f] and RORα[LKO] mice ($n = 6$/group). **i** Macroscopic views of RORα[f/f] and RORα[LKO] mice. **j** Adipose tissues weight of RORα[f/f] and RORα[LKO] mice ($n = 6–7$/group). **k** Representative image of epidydimal white adipose tissue (eWAT) from RORα[f/f] and RORα[LKO] mice stained with hematoxylin and eosin. Scale bar, 100 μm. **l**, **m** Expression levels of inflammatory cytokine genes in eWAT extract **l** or thermogenesis genes in BAT extract **m** from RORα[f/f] and RORα[LKO] mice ($n = 4–5$ per group) as determined by qRT-PCR. Expression was normalized to L32 expression. **n** Metabolic cage studies were performed in RORα[f/f] and RORα[LKO] mice ($n = 5–6$ mice/group). O$_2$ consumption (VO$_2$), CO$_2$ production (VCO$_2$) and energy expenditure were represented (*left to right*). Statistical analysis was performed using Student's unpaired t-test. *$P < 0.05$, **$P < 0.01$, ***$P < 0.001$, NS, Non-Significant. Data expressed as mean ± s.e.m

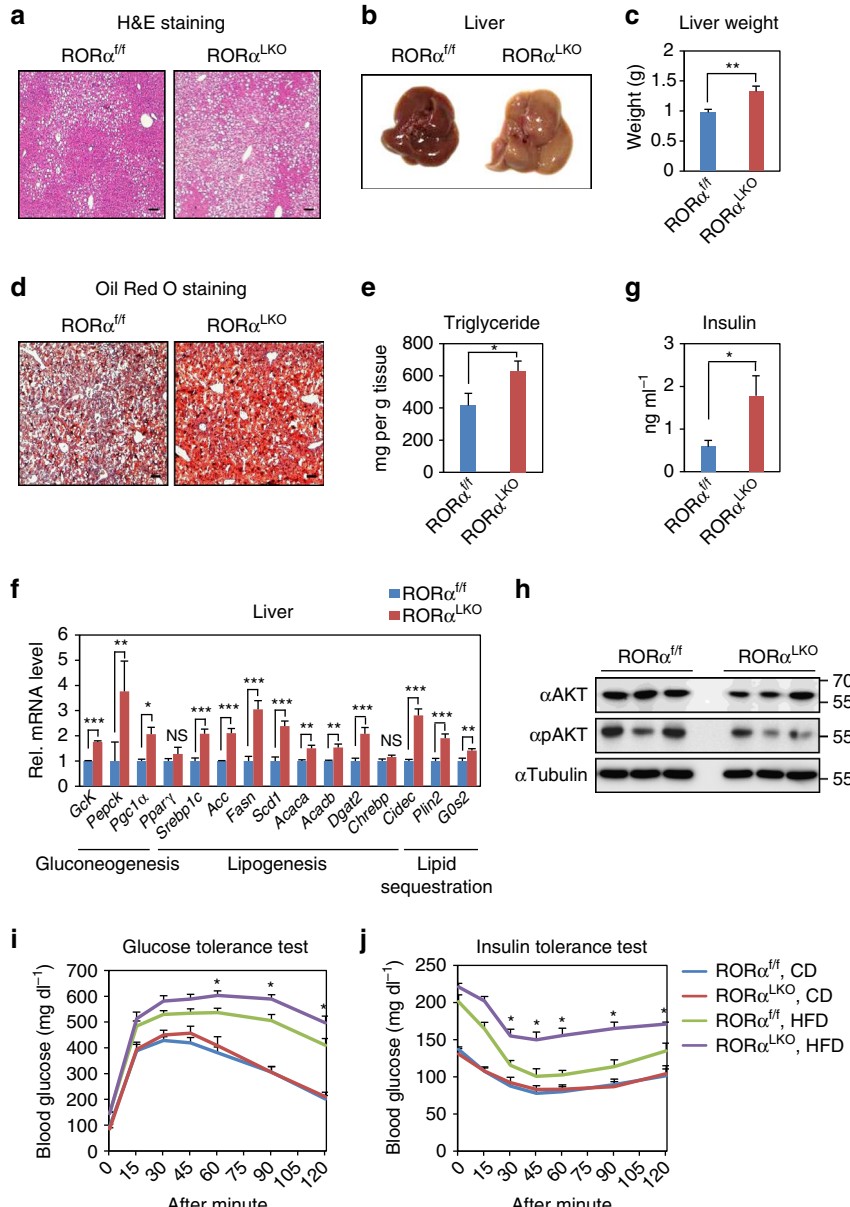

**Fig. 2** Liver-specific *Rorα* deleted mice are susceptible to diet-induced hepatic steatosis and insulin resistance. **a–j** RORα[f/f] and RORα[LKO] mice were fed with HFD for 10 weeks. **a** Representative liver histological section images of RORα[f/f] and RORα[LKO] mice stained with hematoxylin and eosin. Scale bar, 100 μm. **b** Macroscopic view of livers from RORα[f/f] and RORα[LKO] mice. **c** Liver weights of RORα[f/f] and RORα[LKO] mice (n = 10–11 per group). **d** Representative liver histological section images of RORα[f/f] and RORα[LKO] mice stained with Oil Red O. Scale bar, 100 μm. **e** Triglyceride content of livers from RORα[f/f] and RORα[LKO] mice (n = 8 per group). **f** Hepatic gene expression profile involved in metabolism from the livers of RORα[f/f] and RORα[LKO] mice (n = 4 per group) as determined by quantitative PCR with reverese transcription (qRT-PCR). Expression was normalized to 36B4 expression. **g** Fasting insulin levels in RORα[f/f] and RORα[LKO] mice (n = 6–7 per group). Data expressed as mean ± s.e.m. Statistical analysis was performed using Student's unpaired *t*-test. *$P < 0.05$, **$P < 0.01$, ***$P < 0.001$, NS = non-significant. **h** Immunoblot analysis was performed from liver extracts of RORα[f/f] and RORα[LKO] mice. **i, j** Glucose tolerance test **i** and insulin tolerance test **j** on RORα[f/f] and RORα[LKO] mice fed on CD or HFD for 10 weeks. (n = 4–9/group). Data expressed as mean ± s.e.m. Statistical analysis was performed using Student's unpaired *t*-test. *$P < 0.05$, RORα[f/f] vs RORα[LKO], HFD. Data expressed as mean ± s.e.m

dramatically increased PPARγ transcriptional activity, and increased expression of RORα progressively attenuated the PPARγ/PGC1α-dependent transcriptional activation (Fig. 4a). In addition, we examined whether RORα inhibits NCOA1 and NCOA2-mediated PPARγ transcriptional activation. NCOA1 and NCOA2, as p160 family members, are also coactivators for PPARγ[40]. Consistently, RORα significantly reduced NCOA1 and NCOA2-mediated transcriptional activation (Fig. 4b and Supplementary Fig. 3a).

To evaluate the role of RORα in attenuation of the PPARγ-dependent transcriptional activation, we treated Hep3B cells with rosiglitazone, a PPARγ synthetic agonist, and then measured PPRE-luciferase activity. Knockdown of RORα markedly enhanced PPRE-luciferase activity, indicating that RORα functions to repress PPARγ transcriptional activity (Fig. 4c). To determine if DNA-binding domain (DBD) of RORα is required for inhibiting PPARγ transcriptional activation, we introduced DBD-deleted RORα mutant (RORα ΔDBD). We observed

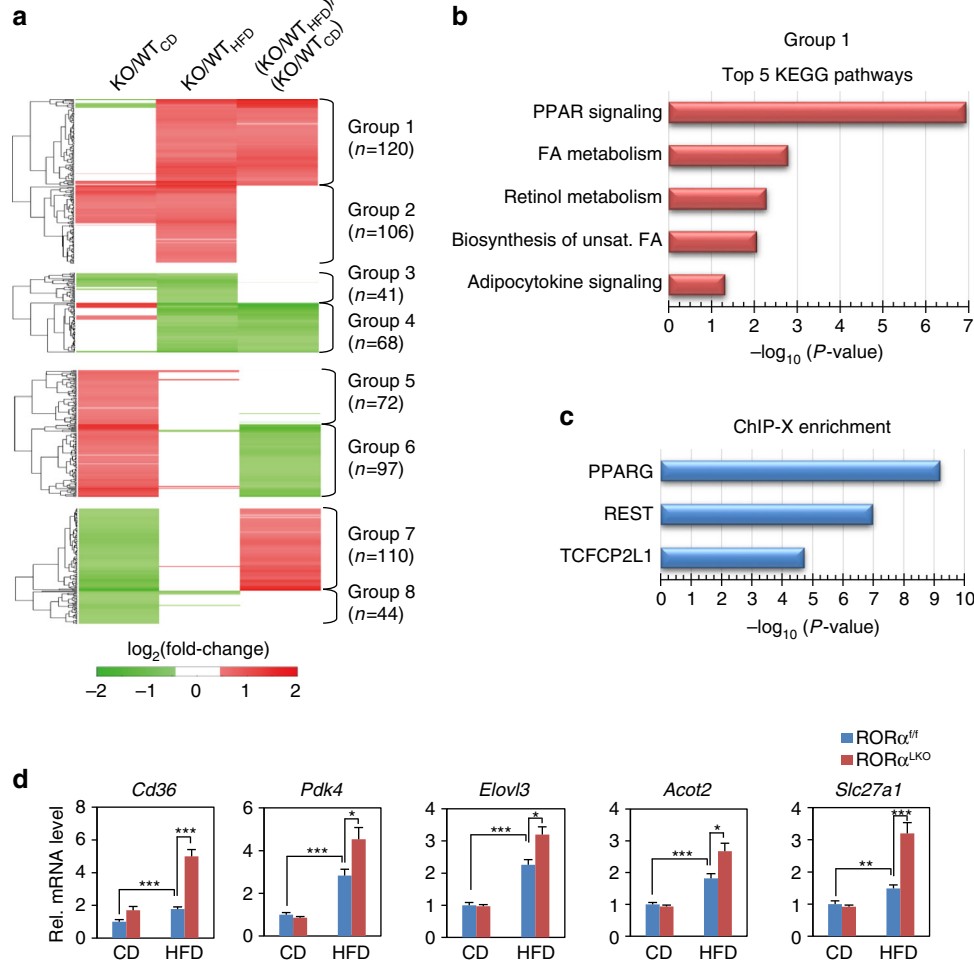

**Fig. 3** Transcriptome analysis of hepatic gene expression profile in RORα$^{LKO}$ mice. **a** Up- and down-regulated genes in RORα$^{LKO}$ compared to RORα$^{f/f}$ mice. These genes were categorized into four groups of the up- (Groups 1, 2) and down-regulated genes (Groups 3, 4) in HFD-fed RORα$^{LKO}$. Besides Groups 1–4, remain genes were also categorized into four groups of the up- (Groups 5, 6) and down-regulated genes (Groups 7, 8) in CD-fed RORα$^{LKO}$. Groups 1, 2 (or Groups 3, 4) were further divided by the specificity of the RORα effect under HFD condition. Log$_2$-fold changes in the following comparisons were displayed: RORα$^{LKO}$/RORα$^{f/f}$$_{HFD}$, RORα$^{LKO}$/RORα$^{f/f}$$_{CD}$ and (RORα$^{LKO}$/RORα$^{f/f}$$_{HFD}$)/(RORα$^{LKO}$/RORα$^{f/f}$$_{CD}$). Hierarchical clustering of the DEGs in Groups 1–8 (Euclidian distance as a dissimilarity measure and average linkage) were used to display the log$_2$-fold changes. **b** KEGG pathways enrichment analysis for the genes in Group 1. The bars represent the enrichment scores, -log$_{10}$ (P value). **c** TF enrichment analysis for the genes in Group 1 using ChEA2 software. Top 3 TFs are shown. The bars represent the enrichment scores, -log$_{10}$ (P value). **d** Expression levels of group 1 genes (upregulated genes in RORα$^{LKO}$ mice fed HFD compared with RORα$^{f/f}$ mice) in liver extract from RORα$^{f/f}$ and RORα$^{LKO}$ mice fed CD or HFD for 10 weeks (n = 5–9/group) as determined by qRT-PCR. Expression was normalized to 36B4 expression. Statistical analysis was performed using two-way ANOVA. *$P < 0.05$, **$P < 0.01$, ***$P < 0.001$. Data expressed as mean ± s.e.m

that RORα WT markedly suppressed PPARγ transcriptional activation, whereas the RORα-mediated repression was remarkably relieved by introduction of RORα ΔDBD (Fig. 4d). As RORα failed to interact with PPARγ, our data proposed that RORα suppresses PPARγ transcriptional activation through DBD and possibly competes with PPARγ for the binding to PPRE. Consistently, the recruitment of RORα was markedly reduced in PPRE-deleted synthetic promoter region (Supplementary Fig. 3b, c).

Since histone acetylation promotes transcriptional activation, we next examined whether RORα interacts with specific histone deacetylases for the repression of PPARγ transcriptional activity. Co-immunoprecipitation assay revealed that RORα specifically interacts with HDAC3 (Fig. 4e and Supplementary Fig. 3d). To determine if HDAC3 is required for RORα-mediated repression of PPARγ transcriptional activity, we further examined repressive function of HDAC3 for PPRE-luciferase activity in the presence or absence of RORα. Intriguingly, knockdown of RORα relieved

the HDAC3-dependent repressive function indicating that HDAC3 exerted repressive function on PPARγ transcriptional activity in the presence of RORα (Fig. 4f). Consistently, knockdown of HDAC3 largely reversed RORα-mediated repression of PPARγ transcriptional activity (Fig. 4g, h and Supplementary Fig. 3e). These results indicate that RORα recruits HDAC3 to potentiate repression of PPARγ transcriptional activity.

**RORα/HDAC3 dynamically regulate PPARγ target gene expression.** PPRE consists of a direct repeat (DR) sequence of (A/G)GGTCA spaced by one nucleotide, whereas consensus RORα response element (RORE) consists of core motif (A/G) GGTCA preceded by a 6-bp A/T-rich sequence. Thus, given that RORE and PPRE share core motif, we hypothesized that RORα directly binds the PPRE of PPARγ target promoters for transcriptional repression. To examine whether RORα and

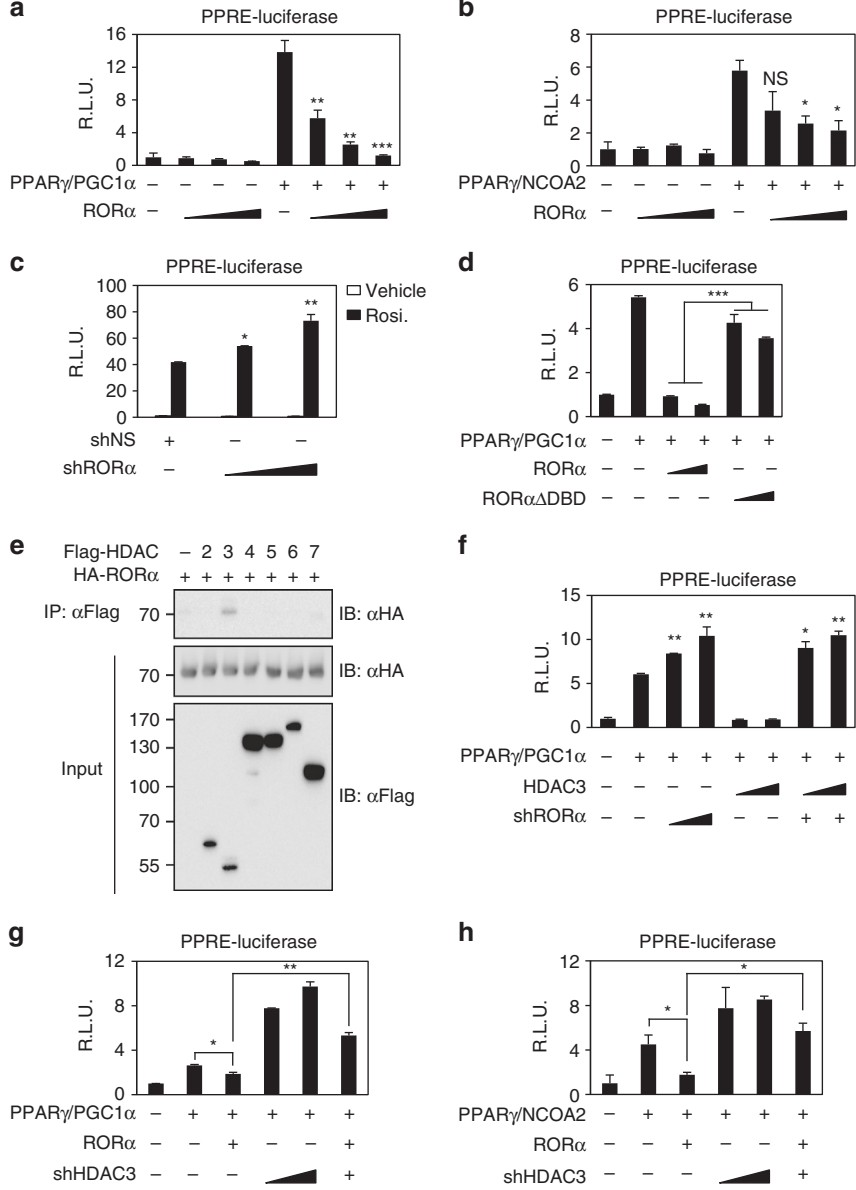

**Fig. 4** RORα interacts with HDAC3 to repress PPARγ transcriptional activity. **a, b** Effect of overexpression of RORα on PPRE-luciferase reporter activity with coactivator PGC1α **a** or NCOA2 **b**. *P < 0.05, **P < 0.01, ***P < 0.001, NS, non-significant, compared to PPARγ/coactivator group. **c** Effect of knockdown of RORα on PPRE-luciferase reporter activity. Cells were treated with DMSO (vehicle), rosiglitazone (20 μM) for 24 h. *P < 0.05, **P < 0.01, compared to shNS group. **d** Effect of RORα ΔDBD mutant on PPRE-luciferase reporter activity. ***P < 0.001. **e** Co-immunoprecipitation assay was performed to detect the interaction between RORα and HDACs of HEK293T cells. **f** Effect of RORα on PPRE-luciferase reporter activity by HDAC3 overexpression. *P < 0.05, **P < 0.01, compared to PPARγ/PGC1α group. **g, h** Effect of knockdown of HDAC3 with coactivator PGC1α **g** and NCOA2 **h** on PPRE-luciferase reporter activity. Data expressed as mean ± s.e.m. Statistical analysis was performed using one-way ANOVA followed by Tukey's post hoc analysis. *P < 0.05, **P < 0.01. Data expressed as mean ± s.e.m

HDAC3 are co-recruited to the PPARγ target promoters for the repression, we performed ChIP assay with anti-RORα, PPARγ, PPARα, RNA polymerase II (Pol II), acetylated H3 (H3Ac) and HDAC3 antibodies from the mouse liver extracts of CD or HFD-fed RORα[f/f] and RORα[LKO] mice. ChIP assays revealed that RORα and HDAC3 were co-recruited to the *Cd36*, *Scd1* and *Plin2* promoters in the liver of HFD-fed RORα[f/f] mice, although no changes were observed with CD-fed RORα[f/f] mice (Fig. 5a and Supplementary Fig. 4a, b). In the absence of RORα, PPARγ recruitment was markedly increased, whereas HDAC3 recruitment was largely diminished along with elevated acetylated H3 levels on the *Cd36*, *Scd1* and *Plin2* promoters in the liver of HFD-fed RORα[LKO] mice (Fig. 5a and Supplementary Fig. 4a, b). Unlike

PPARγ, PPARα recruitment was barely detected from the PPARγ target promoters containing PPRE as assessed by ChIP assay (Fig. 5a and Supplementary Fig. 4b, c).

We next determined if both RORα and HDAC3 are recruited to the PPRE in response to PPARγ agonist in Hep3B cells (Supplementary Fig. 5a). Treatment of rosiglitazone largely induced the expression of PPARγ target genes (Supplementary Fig. 5b). Interestingly, 8 h washout after rosiglitazone treatment dramatically reduced PPARγ target gene expressions (Supplementary Fig. 5a, b). Consistent with gene expressions, treatment of rosiglitazone increased recruitment of PPARγ, PGC1α and Pol II with elevated histone H3 acetylation level on PPARγ target promoters as well as induction of PPARγ target

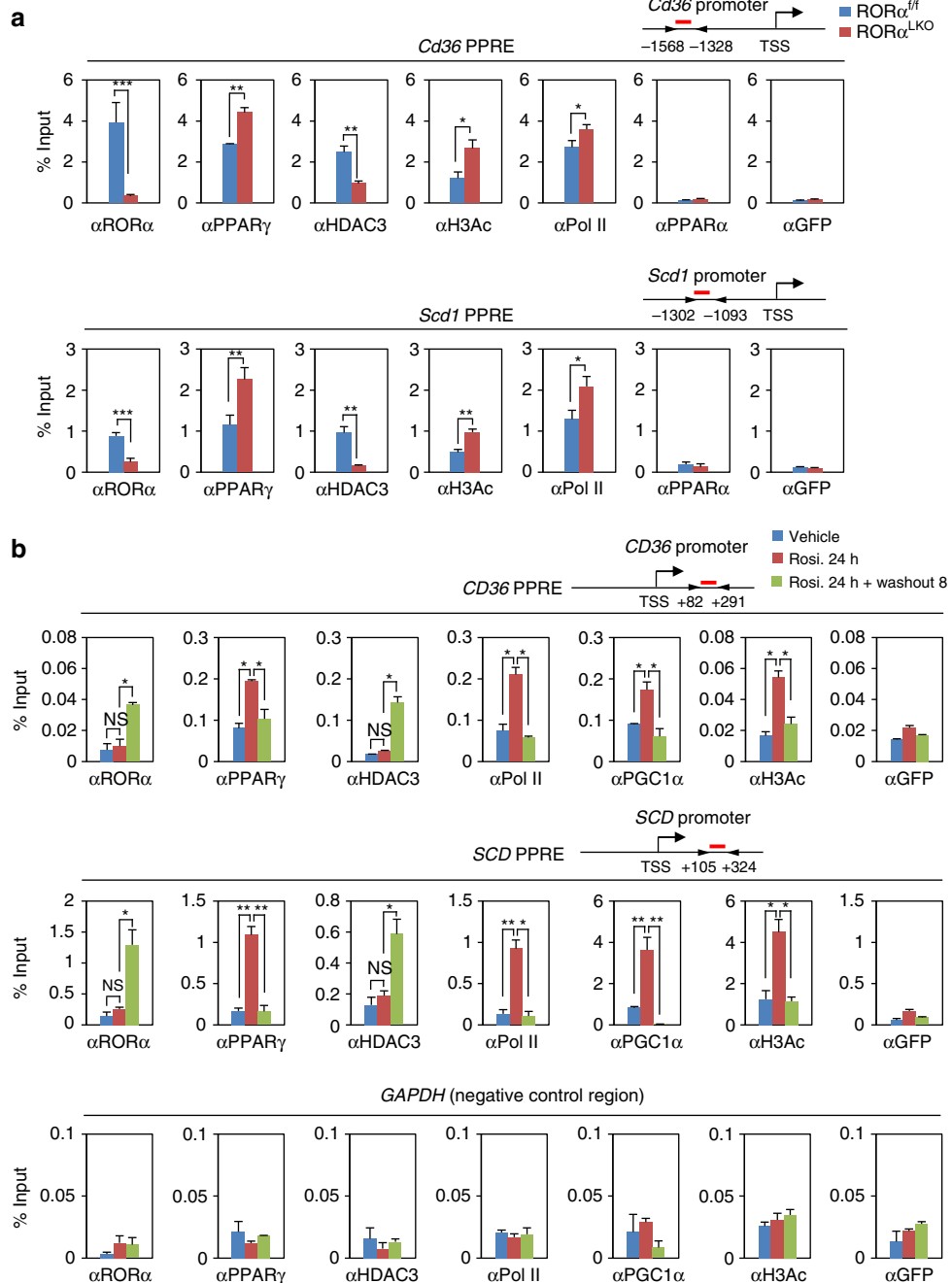

**Fig. 5** RORα recruits to the PPARγ target gene promoters with HDAC3. **a** ChIP assays were performed on the *Cd36* and *Scd1* promoters in liver extract form RORα^{f/f} and RORα^{LKO} mice fed HFD for 10 weeks (*n* = 3 per group). Promoter occupancy by RORα, PPARγ, HDAC3, H3Ac, Pol II, PPARα and GFP was analyzed. Schematic of promoter region was represented with gene name. Red bar depicts locations of PPRE. Statistical analysis was performed using Student's unpaired *t*-test. *P < 0.05, **P < 0.01, ***P < 0.001. **b** ChIP assays were performed on the *CD36, SCD* promoters and *GAPDH*-negative region in Hep3B cells with or without Rosiglitazone (20 μM) treatment for 24 h and washout 8 h. Promoter occupancy of RORα, PPARγ, HDAC3, Pol II, PGC1α, H3Ac and GFP was analyzed. Schematic of promoter region was represented with gene name. Red bar depicts locations of PPRE. Statistical analysis was performed using one-way ANOVA followed by Tukey's post hoc analysis. *P < 0.05, **P < 0.01. Data expressed as mean ± s.e.m

genes (Fig. 5b and Supplementary Fig. 5c). Strikingly, further increased recruitment of RORα to PPRE was observed along with enhanced HDAC3 recruitment in the setting of washout of rosiglitazone for 8 h (Fig. 5b and Supplementary Fig. 5c). Increased recruitment of RORα and HDAC3 substantially diminished PGC1α and Pol II recruitment on PPRE with decreased histone H3 acetylation level on PPRE (Fig. 5b and Supplementary Fig. 5c).

Next, we further determined whether PPARγ antagonist GW9662 also resulted in the increased recruitment of RORα and HDAC3 to the PPARγ target promoters. Consistent with the results from 8 h washout, GW9662 treatment significantly reduced the expression levels of PPARγ target genes (Supplementary Fig. 6a). ChIP assay revealed that recruitment of RORα and HDAC3 to the PPARγ target promoters were markedly increased, while PPARγ and Pol II recruitments were markedly

reduced in response to GW9662 treatment (Supplementary Fig. 6b).

**RORα competes with PPARγ for binding to PPARγ target promoters**. To address HDAC3 recruitment to PPARγ target gene promoters requires RORα, we first examined the PPARγ target gene induction in the presence or absence of RORα. The induction of PPARγ target genes by rosiglitazone was further enhanced by RORα siRNA, indicating that RORα is a critical transcriptional repressor for PPARγ target gene expression (Supplementary Fig. 7a). RORα knockdown largely increased the recruitment of PPARγ for transcriptional activation with increased levels of H3 acetylation to the PPARγ target gene promoters (Fig. 6a and Supplementary Fig. 7b). While remarkably increased by rosiglitazone washout, the HDAC3 recruitment was substantially reduced by RORα knockdown even with setting of rosiglitazone washout (Fig. 6a and Supplementary Fig. 7b). Taken together, these data clearly indicate that RORα is required for recruitment of HDAC3 to PPARγ target gene promoters.

To determine if RORα competes with PPARγ for the binding to the PPARγ target promoters, we transiently knocked down PPARγ in Hep3B cells. To mimic HFD feeding conditions in vitro, we treated cells with free fatty acid (FFA) and examined expression of PPARγ target genes. We observed that FFA treatment markedly increased PPARγ target gene expressions in both WT and *Pparα*-null mouse primary hepatocytes, indicating that PPARα failed to influence on PPARγ transcriptional network in the setting of HFD (Supplementary Fig. 7c). Next, we tested PPARγ target gene expressions in the presence of PPARγ siRNA or HDAC3 siRNA. While repressed by PPARγ siRNA, expression of PPARγ target genes was largely enhanced by HDAC3 siRNA in response to FFA treatment (Supplementary Fig. 7d). Consistent with gene expression, increased PPARγ recruitment by FFA was substantially diminished by PPARγ siRNA (Fig. 6b). Interestingly, recruitment of RORα was dramatically increased to the PPARγ target gene promoters, and HDAC3 recruitment was accompanied by the RORα recruitment to PPARγ target promoters by knockdown of PPARγ (Fig. 6b). Furthermore, Re-ChIP assay clearly indicated that PPARγ and RORα are able to bind to the same genomic region and their recruitments to promoter of target genes are mutually exclusive (Supplementary Fig. 7e). These data strongly indicate that RORα and HDAC3 compete with PPARγ for the binding to the target gene promoters for regulation of gene expressions with opposite transcriptional outputs. Altogether, our data demonstrate that RORα functions as a corepressor along with HDAC3 and is co-recruited to the PPARγ target promoters for the repression of PPARγ-mediated transcriptional activity.

To determine whether RORα recruitment is accompanied by the presence of HDAC3, we examined recruitment of RORα to the PPARγ target gene promoters in the presence of HDAC3 siRNA. Although little or no difference of the recruitment of RORα and PPARγ was observed, Pol II recruitment to the PPARγ target gene promoters was markedly increased by HDAC3 knockdown (Fig. 6c), indicating that the presence of HDAC3 affected the recruitment of RNA polymerase II to the PPARγ target gene promoters. Taken together, these data indicate that both RORα and HDAC3 serve as transcriptional corepressors on the PPARγ target gene promoters for the repression of PPARγ target gene expressions.

**PPARγ antagonism restores metabolic homeostasis in RORα^LKO mice**. Since RORα turned out to play a key role to repress PPARγ transcriptional activity in vitro and in vivo, we next examined if inhibition of PPARγ transcriptional activities restores impaired metabolic homeostasis. For this, PPARγ antagonist GW9662 were treated to RORα^f/f and RORα^LKO mice for 5 weeks with HFD. Intriguingly, the body weight gain of both RORα^f/f and RORα^LKO mice were markedly reduced by GW9662 treatment compared with vehicle-treated control mice (Fig. 7a). The reduction of body weight gain by GW9662 in RORα^LKO mice was much greater, leading to similar body weight to GW9662-treated RORα^f/f mice, indicating that inhibition of PPARγ activity remarkably reduces body weight gain in RORα^LKO mice (Fig. 7a). Similar to reduced body weight gain, tissue weights of the liver and epididymal white adipose tissue were markedly reduced by GW9662 treatment in both RORα^f/f and RORα^LKO mice (Fig. 7b, c). Consistently, cross-sectional area of adipocytes was significantly reduced by GW9662 treatment (Fig. 7d). In accordance with body weight reduction, PPARγ antagonism markedly reduced hepatic steatosis in both RORα^f/f and RORα^LKO mice (Fig. 7e). Consistently, gene expression profile analysis exhibited that target gene expression levels involved in hepatic gluconeogenesis and lipogenesis are largely reduced by GW9662 treatment (Fig. 7f, g). Together, we demonstrate that enhanced PPARγ transcriptional activity by RORα deficiency is de-activated by PPARγ antagonism to restore metabolic homeostasis, including body weight gain, hepatic steatosis and glucose and lipid metabolism.

## Discussion

Hepatic nuclear receptors play critical roles in the regulation of lipid and glucose metabolism in response to environmental stress, including nutrient and hormonal cues[41]. Dysfunction of hepatic nuclear receptors is largely linked to metabolic diseases including obesity and type II diabetes. We found that PPARγ signaling is a critical pathway affected by hepatic deletion of RORα. Dysregulated PPARγ signaling in RORα^LKO mice results in uncontrolled lipogenesis, contributing to the development of hepatic steatosis and diet-induced obesity on a HFD. Furthermore, treatment of PPARγ antagonist GW9662 decreased the susceptibility to obesity[42, 43]. Consistent with previous reports, we also observed that elevated PPARγ transcriptional activity in RORα^LKO mice are downregulated after treatment of GW9662, resulting in decrease of diet-induced hepatic steatosis and obesity. Our data confirm that RORα is a key factor for the repression of PPARγ signaling to protect against diet-induced hepatic steatosis and obesity in vivo.

Together, PPARγ signaling turns out to be significantly activated in HFD-fed RORα^LKO mice while increased RORα reduces PPARγ transcriptional activity, providing a direct molecular link between RORα and PPARγ. Furthermore, our data indicate that RORα regulates PPARγ signaling through RORα-mediated HDAC3 recruitment to the PPARγ target promoters. Thus, RORα plays a crucial role in maintaining homeostasis of lipid metabolism in liver by negatively regulating PPARγ signaling via HDAC3 recruitment to the PPARγ target promoters for transcriptional repression (Fig. 7h).

Thiazolidinedione (TZD) is a synthetic PPARγ agonist and has been clinically approved to improve glucose homeostasis and fatty liver in human patients. Although the molecular mechanisms still remain unclear, the 'lipid steal' hypothesis has been widely accepted to explain of how TZD treatment improves insulin resistance in type II diabetes patients[44–46]. However, though PPARγ activation has shown to reduce blood glucose level and hepatic gluconeogenesis, and improve glucose tolerance[47, 48], several reports have shown that PPARγ activation leads to hepatic steatosis[49, 50]. In general, the expression of PPARγ is very low in human and mouse liver. Interestingly, the expression level of hepatic PPARγ is significantly upregulated in obese rodent model[51] and high level of PPARγ in mouse liver is sufficient for

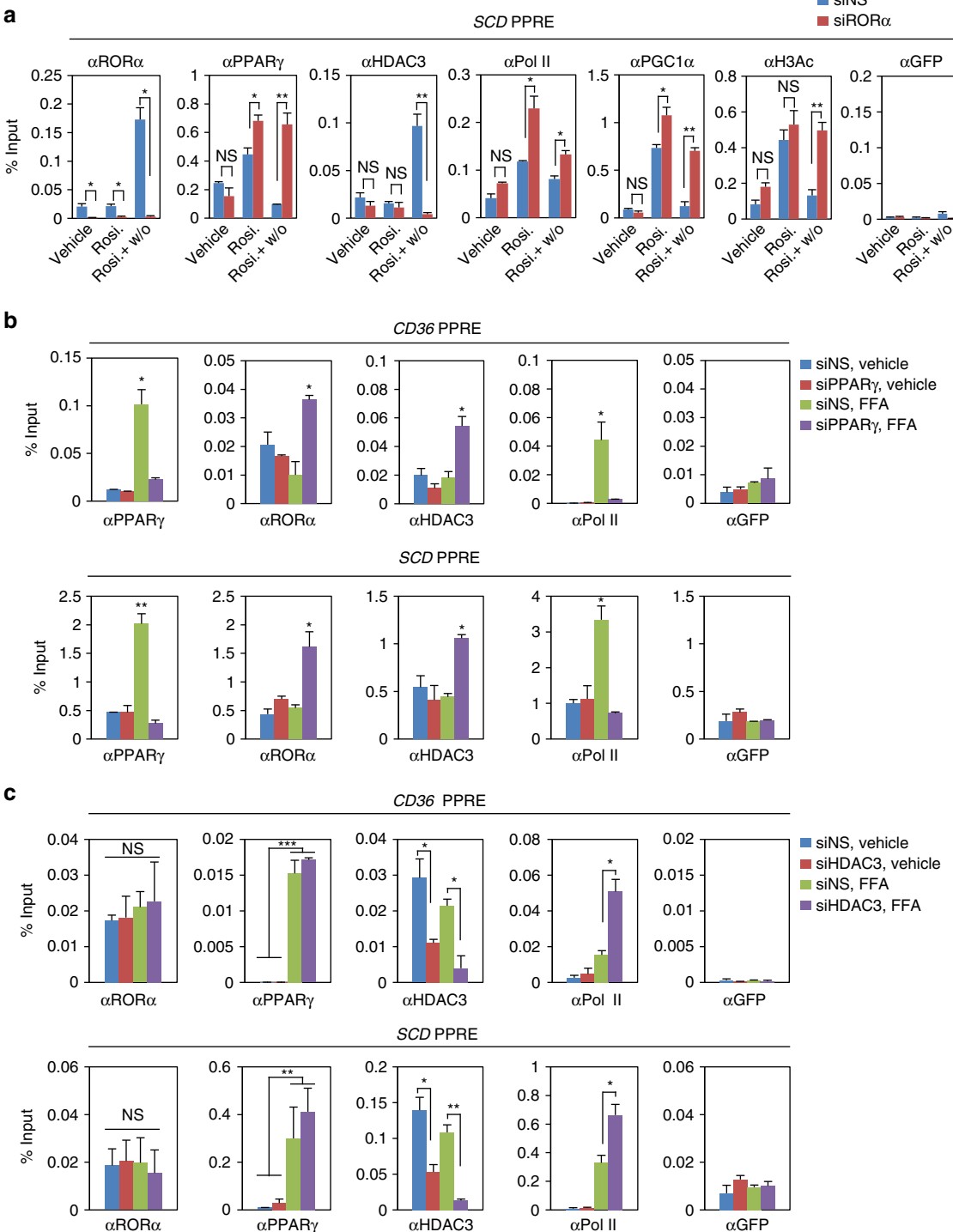

**Fig. 6** Recruitment of RORα and PPARγ to the PPARγ target gene promoters are mutually exclusive. **a** ChIP assays were performed in the absence or presence of RORα on *SCD* promoters in Hep3B cells with or without Rosiglitazone (20 μM) treatment for 24 h and washout 8 h. Promoter occupancy of RORα, PPARγ, HDAC3, Pol II, PGC1α, H3Ac and GFP was analyzed. Statistical analysis was performed using Student's unpaired *t*-test. *P < 0.05, **P < 0.01, NS, non-significant. **b**, **c** ChIP assays were performed in the absence or presence of PPARγ **b**/HDAC3 **c** on the *CD36* and *SCD* promoters in Hep3B cells with or without free fatty acid (free fatty acid: Oleic acid 200 μM and Palmitic acid 100 μM) treatment for 24 h. Promoter occupancy of PPARγ, RORα, HDAC3, Pol II and GFP was analyzed. Data expressed as mean ± s.e.m. Statistical analysis was performed using one-way ANOVA. *P < 0.05, **P < 0.01, ***P < 0.001. Data expressed as mean ± s.e.m

the induction of adipogenic transformation of hepatocytes with adipose tissue-specific gene expression and lipid accumulation[52]. These data indicate that PPARγ plays a key role in development of hepatic steatosis. Accordingly, inhibition of PPARγ signaling and hepatic deficiency of PPARγ in ob/ob mice have shown to

ameliorate fatty liver[53, 54]. A recent study clearly showed that PPARγ antagonism improves insulin sensitivity, promotes the browning of white adipose tissue and reduces lipogenic and glucogenic gene expressions in the liver to prevent against diet-induced obesity[55]. Intriguingly, liver-specific PPARγ-

deficient mice exhibit resistance to HFD-induced hepatic steatosis[56]. Expression of numerous genes involved in lipid uptake and lipid transport was remarkably decreased in the liver-specific PPARγ-deficient mice, resulting in reduction of hepatic steatosis[56]. Altogether, these studies suggest that local activation of hepatic PPARγ may promote ectopic fat deposition in the liver whereas systemic activation of PPARγ may promote fat deposition in adipose tissue rather than liver leading to clinical improvements of metabolic syndromes including hepatic

steatosis. Thus, negative control of RORα to suppress hepatic PPARγ activation is important to maintain physiological hepatic lipid homeostasis. Therefore, these results strongly indicate that the PPARγ signaling pathway is involved in diet-induced hepatic steatosis, and hepatic lipid accumulation is prevented by suppression of PPARγ transcriptional network in the liver.

It has been widely accepted that PPARα is a major nutrient-sensing PPAR isoform to modulate hepatic gene expressions[37]. As no substantial activation of PPARα has been

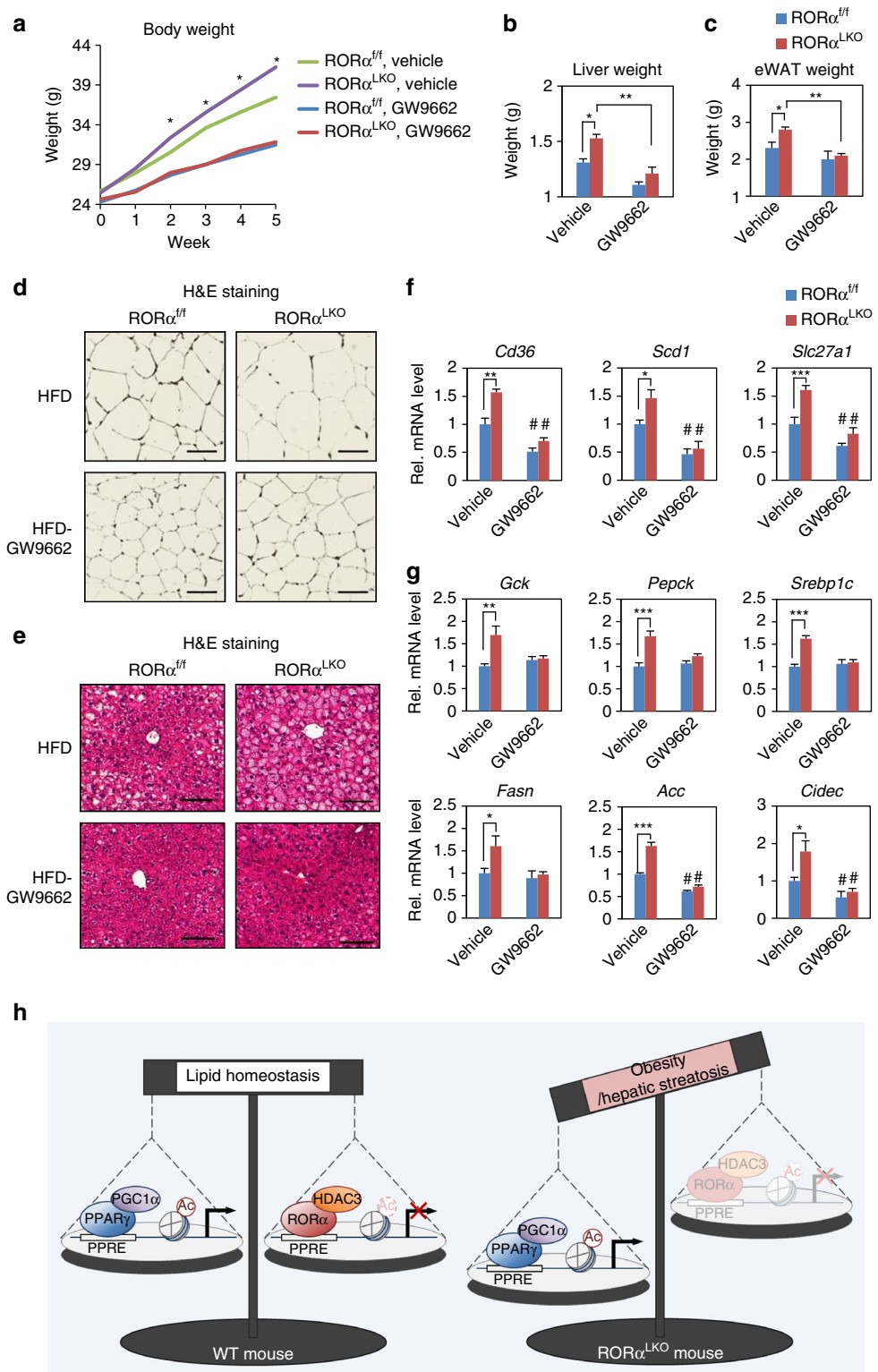

observed in the fasted or HFD-fed RORα[LKO] mice, we believe that RORα mainly controls PPARγ transcriptional network to maintain hepatic homeostasis in response to HFD. However, it has been well established that PPARα is a promising therapeutic target to upregulate beta oxidation gene expressions and inhibit hepatic de novo lipogenesis. Consistently, animal model using PPARα-null mice have been reported to develop remarkable hepatic steatosis[38, 57]. Thus, it is highly possible that hepatic steatosis phenotype in HFD-fed RORα[LKO] mice may be resulted from both upregulated PPARγ activation and suppressed PPARα transcriptional activity in the absence of RORα. Understanding of the contribution of PPARα transcriptional network with RORα would be critical to delineate the molecular mechanisms of how PPAR isoforms including PPARα and PPARγ modulate hepatic lipid homeostasis with various transcriptional factors in response to environmental stress such as HFD.

It has been reported that RORα may compete with PPARγ for the binding to PPRE[58]. It is well established that PPRE contains a DR1 motif consisting of two core DRs of AGGTCA separated by a single nucleotide[59]. Among nucleotides of DR1 motif, the four nucleotides immediately 5′ of DR1 motif are highly conserved and exhibit a consensus of A(A/T)CT. Previous study has reported that the binding of the DBD of PPARs to the single core binding site requires the AT-rich 5′-extended binding site which is quite similar to the binding site for the monomer of RORα[60]. Thus, the similarity in the binding sequences for PPARγ and RORα appears to allow RORα to modulate PPAR signaling by competing with PPARγ for binding to PPREs[61].

The physiological role of HDAC3 has been reported to repress hepatic steatosis. In liver-specific *Hdac3*-deficient mice, little or no body weight change was observed. As HDAC3 regulates the expression of lipogenic genes in an enzymatic activity-independent manner[62], fasting phase markedly promotes hepatic steatosis in liver-specific *Hdac3*-deficient mice[63]. An intriguing observation in this study is that RORα is crucial to recruit HDAC3 to repress hepatic PPARγ-mediated lipogenic genes and protect against diet-induced hepatic steatosis and obesity. Furthermore, repressive role of RORα-mediated HDAC3 on lipid metabolism is coupled with elevated hepatic gluconeogenesis. Though hepatic HDAC3 has been shown to promote gluconeogenesis by repressing lipid synthesis and sequestration[63], we observed notable increase of gene expression involved in hepatic gluconeogenesis in HFD-fed RORα[LKO] mice. Intriguingly, HDAC3 ablation upregulated hepatic expression of perilipin gene which contributes to lipid sequestration to ameliorate glucose tolerance[63]. We also noticed that perilipin 2, hepatic isoform of perilipin was substantially elevated in HFD-fed RORα[LKO] mice. Unlike hepatic HDAC3 ablation, impaired RORα-mediated HDAC3 transcriptional repression led to interfere hepatic homeostasis of PPARγ signaling. Therefore, disturbed regulatory mechanism of PPARγ signaling in HFD-fed RORα[LKO] mice would be the main cause of the insulin resistance

and glucose intolerance. Consistent with elevated fasting glucose level in HFD-fed RORα[LKO] mice, mRNA level of the rate-limiting enzyme, phosphoenol pyruvate carboxykinase (PEPCK), in the hepatic gluconeogenesis pathway, was largely elevated in HFD-fed RORα[LKO] mice. Together, our data strongly indicate that physiological role of HDAC3 in the liver is to suppress PPARγ transcriptional activity via RORα to control hepatic lipid and glucose metabolism.

Previously, it has been reported that bile acid signaling pathway is critical to modulate EE in brown adipose tissue. Bile acids activates mitogen-activated protein kinase pathways and serve as ligands for the G protein-coupled receptor TGR5[29]. Thus, hepatic bile acid synthesis and bile acid pool size in the serum is critical to control metabolic rate. Bile acids induces cyclic-AMP-dependent thyroid hormone activating enzyme type 2 iodothyronine deiodinase (D2)[29]. Thus, bile acid-TGR5-cAMP-D2 signaling pathway in the brown adipose tissue has been known as a crucial mechanism to modulate EE[29]. We observed that several key genes involved in bile acid synthesis were largely downregulated as well as serum bile acid pool size in HFD-fed RORα[LKO] mice. Though we still do not know the direct mechanism of how hepatic bile acid signaling was impaired in RORα[LKO] mice, we speculate that impaired hepatic bile acid synthesis would impair TGR5 activation in brown adipose tissue to reduce EE.

Several of the observed metabolic alterations in the RORα[LKO] mice are indeed different from those observed in *sg* mice. For example, RORα[LKO] mice gain significantly more weight than WT mice and develop hepatic steatosis when fed with HFD. However, *sg* mice are protected from HFD-induced obesity and fatty liver and display improved insulin sensitivity[28, 64]. A strong difference between these two mouse models is their overall growth condition. RORα[LKO] mice have no obvious phenotypic abnormalities under normal dietary conditions, whereas *sg* mice suffer from severe growth retardation that would likely be attributed to a number of developmental defects. In addition, defective RORα function in other tissues including brain is likely to systemically affect energy intake and expenditure in the *sg* mice, making it difficult to specifically dissect hepatic function of RORα. Therefore, it would be helpful to utilize these two mice and compare their phenotypes in certain conditions together for understanding of RORα function *in vivo*. Collectively, our data indicate that liver-specific *RORα* deficient mice were successfully developed and the utilization of the mice allowed us to be able to study *in vivo* functions of RORα in liver more precisely by excluding the potential secondary effect of *sg* mice.

In conclusion, our data indicate that RORα requires HDAC3 to regulate PPARγ signaling to maintain lipid homeostasis in response to over-nutrient cue. We demonstrate that major target of RORα in the liver is the PPARγ signaling and lipid/glucose metabolism. Our findings provide a direct link between RORα and hepatic fatty acid and glucose metabolism. Thus, therapeutic

**Fig. 7** PPARγ antagonism restores metabolic homeostasis in RORα[LKO] mice. **a–e** RORα[f/f] and RORα[LKO] mice were fed HFD with or without GW9662 for 5 weeks ($n = 4$–5 per group). **a** Body weight curves. Statistical analysis was performed using Student's unpaired *t*-test. *$P < 0.05$, RORα[f/f] vs RORα[LKO], vehicle. **b, c** Liver **b** and epididymal white adipose tissue (eWAT) **c** weight of RORα[f/f] and RORα[LKO] mice were fed HFD with or without GW9662 for 5 weeks. Statistical analysis was performed using two-way ANOVA. *$P < 0.05$, **$P < 0.01$. **d, e** Representative histological section images from eWAT **d** and liver **e** of RORα[f/f] and RORα[LKO] mice fed HFD with or without GW9662 for 5 weeks. Scale bar, 100 μm. **f, g** Expression levels of PPARγ target genes **f** or gluconeogenesis/lipogenesis/lipid sequestration genes **g** in liver from RORα[f/f] and RORα[LKO] mice fed HFD with or without GW9662 for 5 weeks as determined by quantitative PCR with reverese transcription. Expression was normalized to 36B4 expression. Statistical analysis was performed using Student's unpaired t-test. *$P < 0.05$, **$P < 0.01$, ***$P < 0.001$, #$P < 0.05$ compared to each vehicle group. Data expressed as mean ± s.e.m. **h** Proposed model for the role of RORα in hepatocyte. RORα regulates PPARγ signaling via HDAC3 recruitment to the PPARγ target promoters for transcriptional repression

strategies designed to modulate RORα activity may be beneficial for the treatment of hepatic disease as well as obesity-associated metabolic diseases.

## Methods

**Generation of conditional Rorα-deficient mice and animal care.** To generate mice with a floxed RORα allele, a 16.5 kb region used to construct the targeting vector was first subcloned from a BAC clone (bMQ-293I20, Source BioScience) into a pBluescript phagemid system. The FRT-flanked puromycin cassette containing a loxP sequence was inserted at the front of exon 4 and the single loxP site was inserted at the back of exon 5. The target region was ~15.2 kb which included exons 4 and 5. Twenty micrograms of the targeting vector was linearized by SalI and then electroporated to E14Tg2A ES cells. Surviving clones after puromycin selection were expanded and analyzed by Southern blot to confirm recombinant ES clones. After BamHI digestion, the bands representing WT and mutant alleles are 9.0 and 6.8 kb, respectively. Targeted ES cells were selected for microinjection into C57BL/6 blastocysts to generate chimeras. The male chimeras were bred with C57BL/6 female mice to select for germline transmission. To remove the puromycin selection cassette, targeted heterozygous F1 was crossed with Flp deleter strain (FLPeR mice, The Jackson Laboratory strain 003946). The mice were backcrossed to C57BL/6 then crossed with Alb-Cre mice (The Jackson Laboratory strain 003574) to generate liver-specific Rorα-deficient mice. Male RORα$^{f/f}$ and RORα$^{LKO}$ mice at 8 weeks of age were fed a CD or a 60% kcal fat HFD (Research diet, D12492) during 10 weeks. The sample sizes for all animal studies were announced in each figure legend. Mice were housed in a specific pathogen-free AAALAC-accredited facility under controlled conditions of temperature (25 °C) and light (12 h light:12 h dark, lights switched on at 7:00 a.m.). Food and water were available ad libitum. All mice used in these experiments were backcrossed to C57BL/6 at least seven generations. Animals for each group of experiments were chosen randomly. The primers used in PCR analysis for genotyping floxed alleles are: forward 5′-GCTTGTGGGTTTCTCCTACA-3′ and reverse 5′-GCAGCAAGTGTTGTGTCCCA-3′. This study was reviewed and approved by the Institutional Animal Care and Use Committee (IACUC) of National Cancer Center Research Institute.

**Body composition.** Fat and lean body masses were assessed by $^1$H magnetic resonance spectroscopy (Bruker BioSpin).

**Indirect calorimetry.** Oxygen consumption (VO$_2$), carbon dioxide production (VCO$_2$), respiratory exchange ratios, EE and food consumption were measured using an indirect calorimetry system PHENOMASTER (TSE System). Mice in each chamber were maintained at a constant environmental temperature of 22 °C.

**Isolation and culture of primary mouse hepatocytes.** Mouse primary hepatocytes were isolated from the liver of 8-week-old male RORα$^{f/f}$ and RORα$^{LKO}$ mice or WT and PPARα null mice by the collagenase perfusion method[65]. Dissociation into individual hepatocytes was performed in Dulbecco's modified Eagles' medium (DMEM) (Welgene) containing 10% heat-inactivated fetal bovine serum (FBS), 1% antibiotics, 20 mM HEPES, 100 nM insulin, 1 nM dexamethasone. For each hepatocyte preparation, cell viability was estimated by the exclusion of trypan blue.

**Total bile acid (TBA) measurement.** The quantitative determination of total bile acid of mice serum that was collected after centrifugation of mice blood was measured using the total bile acids assay kit (DZ042A-K, Diazyme Laboratories), according to the manufacturer's instructions.

**Histology.** When mice were euthanized by CO$_2$ asphyxiation, livers and white adipose tissues (WATs) were rapidly fixed in 10% formalin (Sigma) at 4 °C overnight. After fixation, tissues were sequentially dehydrated in ethanol with increasing concentrations ranging from 50 to 100%. Dehydrated specimens were subsequently infiltrated with 100% xylene and embedded in paraffin wax. For hematoxylin and eosin (H&E) staining, tissues were sectioned at 5 μm thickness, deparaffinized, rehydrated and stained with hematoxylin for 3 min followed by counterstaining with eosin for 1 min. For Oil red O staining, fresh samples of liver embedded in OCT tissue freezing medium (Sakura Finetek). 0.5% Oil red O solution was prepared by dissolving 0.5 g Oil red O powder (Sigma) in 100 ml propylene glycol (sigma). Fresh frozen specimens were cryosectioned at 8 μm thickness and air dried. Then fix in ice cold 10% formalin for 10 min, air dried again, and rinsed with distilled water. Sections were placed in 100% propylene glycol for 5 min and stained with pre-warmed 0.5% Oil red O solution in propylene glycol for 15 min in 60 °C oven. Then sections were rinsed with distilled water and followed by counterstaining with hematoxylin. Images were acquired using digital microscopes (Leica DMD108, Leica microsystems) equipped with ×10 and ×20 objective lenses.

**Quantitative real-time RT-PCR.** Total RNAs were extracted using Trizol (Invitrogen) and reverse transcription was performed from 2.5 μg of total RNAs using the M-MLV cDNA Synthesis kit (Enzynomics). The abundance of mRNA was detected by a CFX384 Touch$^{TM}$ Real-Time PCR Detection System (Bio-Rad) with SYBR Green (Enzynomics). The quantity of mRNA was calculated using ΔΔCt method and normalized by using primers indicated in each figure legend. All reactions were performed as triplicates. Primers used for analysis are listed in Supplementary Data 5.

**Intraperitoneal glucose or insulin tolerance tests.** For GTTs, 2 g of glucose per kg of mice body weight was injected i.p. to overnight fasted mice. For ITTs, 0.75 U of insulin (Humulin R, Eli Lilly) per kg of mice body weight was injected i.p. to 6 h fasted mice. Mice blood was drawn at indicated time intervals from the tail tip puncture, and blood glucose level was measured by accu-check perfoma glucometer (Roche).

**Generation of mRNA-sequencing data.** Four groups of mice, CD-fed RORα$^{f/f}$, CD-fed RORα$^{LKO}$, HFD-fed RORα$^{f/f}$ and HFD-fed RORα$^{LKO}$ mice, were analyzed by RNA-sequencing. Four mice per group were killed, and livers from two mice were pooled to generate two samples per group, that is, duplicate experiments for each group were performed. Total RNA extraction was performed using Trizol (Invitrogen). Poly(A) mRNA isolation from total RNA and fragmentation were performed using the Illumina Truseq RNA Sample Prep Kit with poly-T oligo-attached magnetic beads, according to the manufacturer's instructions. Reverse transcription of RNA fragments was performed using Superscript II reverse transcriptase (Life Technologies). The adaptor-ligated library was size-selected by band excision after agarose gel electrophoresis and purified using the QIAquick gel extraction kit (Qiagen). The prepared mRNA-sequencing libraries were pair-end sequenced on an Illumina Hi-seq 2500. The accession number for the mRNA-sequencing data in this paper is GSE83338.

**Analysis of mRNA-sequencing data.** After removing adapter sequences (TrueSeq universal and index adapters), we used cutadapter software[66] to trim the reads that PHRED scores lower than 20. Remaining reads were aligned to the mouse reference genome (GRCm38) using TopHat aligner[67]. After the alignment, we quantified the expression of genes as Fragments Per Kilobase of transcript per Million mapped reads (FPKM) for each gene using Cufflinks[68]. To identify the DEGs, we first selected the 'expressed' genes as the ones with FPKM larger than 1 under at least one of the eight samples. For the expressed genes, log$_2$(FPKM + 1) values were normalized across eight samples using the quantile normalization method. To identify the DEGs, for each gene, we calculated a T-statistic and log$_2$-fold-change in the comparisons of RORα$^{LKO}$/RORα$^{f/f}$$_{HFD}$ and RORα$^{LKO}$/RORα$^{f/f}$$_{CD}$. We then estimated empirical distributions of T-statistics and log$_2$-fold changes for the null hypothesis by random permutation of the eight samples (1000 permutations). On the basis of the distributions, for each gene, we computed adjusted P values for the observed T-statistic and log$_2$-fold-change and the combined these P values with Stouffer's method[69]. Finally, we identified the DEGs as the ones that have the combined P-value ≤ 0.05 and absolute log$_2$-fold-change ≥ 0.439, which is a cutoff value (the 95th percentile of the empirical distribution for log$_2$-fold changes) for each comparison. We further identified RORα-dependent genes under HFD condition as the ones with significant differences between the log$_2$-fold-change in the two comparisons above (RORα$^{LKO}$/RORα$^{f/f}$$_{HFD}$ and RORα$^{LKO}$/RORα$^{f/f}$$_{CD}$) larger than 0.439.

**Functional enrichment analysis and TF enrichment analysis.** For the genes in Groups 1–8, enrichment analysis of GOBPs and KEGG pathways were performed using a DAVID software[34]. We selected the GOBPs and KEGG pathways with P-value < 0.05 as the ones represented by the genes analyzed. For the genes in Group 1, TF enrichment analysis was performed using a ChEA2 software[33]. Among the TF-target gene data, only mouse TF-target gene data were used for the enrichment analysis. We selected the TFs with P-value < 0.01 as the ones significantly regulating the genes in Group 1.

**Luciferase reporter assay.** HEK293T cells (ATCC) and Hep3B cells (Korean Cell Line Bank) were grown and transiently transfected by using polyethylenimine (PEI) and turbofect (Thermo Scientific, R0531). All cell lines used in the study were regularly tested for mycoplasma contamination. For luciferase reporter assays, 1 × 10$^5$ cells were seeded in DMEM supplemented with 10% FBS and 1% antibiotics. Cells were transfected with PPRE-luciferase reporters and β-galactosidase expression constructs along with several expression constructs were indicated in each figure. Using a luciferase assay system (Promega), the luciferase activity was measured with a luminometer (Berthold Technologies) after 48 h of transfection. Transfection efficiency was normalized by β-galactosidase expression. The results were obtained from at least three independent experiments

**Co-immunoprecipitation assay.** HEK293T cells that transfected with Flag-HDACs and HA-RORα were cultured and lysed with lysis buffer (200 mM NaCl, 50 mM Tris-HCl, pH 8.0 and 0.5% NP40). About 20 mg of cell extracts was

immunoprecipitated with each 1 µg of anti-Flag antibody overnight and then incubated with 35 µl (50% slurry) of protein A/G agarose beads for 1 h. The immunoprecipitated materials were washed with 500 µl of washing buffer (150 mM NaCl, 50 mM Tris-HCl, pH 8.0 and 0.5% NP40) for four times and bound materials were eluted by boiling in 50 µl of sampling buffer (2% β-mercaptoethanol, 5% glycerol, 1% SDS and 60 mM Tris-HCl, pH 6.8) and subjected to immunoblot analysis. Protein samples were resolved by sodium dodecyl sulphate-polyacrylamide gel electrophoresis Images of the immunoblots were visualized and recorded using the LAS 4000-mini system (Fujifilm). Original uncropped images of immunoblots used in this study can be found in Supplementary Fig. 8.

**Chromatin Immunoprecipitation (ChIP) and Re-ChIP assays.** The ChIP assays were conducted as described. Cells were crosslinked with 1% formaldehyde for 10 min at room temperature. Mouse livers were harvested and quickly washed with PBS and crosslinked with 1% formaldehyde for 10 min at room temperature, followed by quenching with 0.125 M glycine solution for 5 min. Then, cells or harvested mouse livers were washed with ice-cold PBS two times. Chromatin fragmentation was performed by sonication in ChIP lysis buffer (50 mM Tris-HCl (pH 8.1), 1% SDS, 10 mM EDTA (pH 7.6), and protease inhibitor cocktail) with an average size of approximately 500 bp. Proteins were immunoprecipitated in ChIP dilution buffer (1% Triton X-100, 2 mM EDTA, 150 mM NaCl, 20 mM Tris-HCl (pH 8.1), and protease inhibitor cocktail). Crosslinking was reversed overnight at 65 °C in elution buffer (1% SDS, 0.1 M NaHCO₃), and DNA was purified with a QIAquick Gel extraction Kit (QIAGEN). For the Re-ChIP assay, components were eluted from the first immunoprecipitation reaction by incubation with 10 mM DTT at 37 °C for 30 min and diluted 1:50 with ChIP dilution buffer containing 20 mM Tris-HCl (pH 8.1), 150 mM NaCl, 2 mM EDTA, and 1% Triton X-100 followed by reimmunoprecipitation with the secondary antibody. Precipitated DNA was analyzed by quantitative PCR. For real-time quantitative PCR analysis, 2 µl from 60 µl DNA extractions was used. All reactions were performed in triplicates. Primers used for analysis are listed in Supplementary Data 5.

**GW9662-treated mice.** RORα$^{f/f}$ and RORα$^{LKO}$ male mice at 8 weeks of age were subjected to GW9662 at a dose of 0.35 mg per kg body weight per day or an equivalence volume of vehicle in their drinking water for 5 weeks with feeding HFD. The sample sizes for this study was announced in figure legend.

**Antibodies.** Commercially available antibodies were used: anti-RORα (sc-28612; 1:1000 dilution for IB analysis, 5 µg for ChIP assay), anti-tubulin (sc-8035, 1:5000 dilution for IB analysis), anti-AKT (sc-8312; 1:1000 dilution for IB analysis), anti-PPARα (sc-9000x, 1 µg for ChIP assay) and anti-GFP (sc-9996, 1 µg for ChIP assay) from Santa Cruz Biotechnology; anti-β-actin (A5441; 1:5000 dilution for IB analysis) and anti-FLAG (F3165, Sigma, 1:5000 dilution for IB analysis, 1 µg for IP assay) from Sigma; anti-HA (MMS-101R; 1:5000 dilution for IB analysis, 1 µg for IP assay) from Covance; anti-H3Ac (#06-599, 1 µg for ChIP assay) from Millipore; anti-phospho-AKT(Ser473) (#4051 S, 1:1000 dilution for IB analysis) from Cell Signaling; anti-PPARγ (ab41928, 1 µg for ChIP assay), anti-PGC1α (ab54481, 1 µg for ChIP assay) and anti-HDAC3 (ab7030, 1 µg for ChIP assay) from Abcam; anti-RNA polymerase II (MMS-126R, 1 µg for ChIP assay) from Berkeley antibody company; anti-V5 (R96025; 1:5000 dilution for IB analysis) from Invitrogen.

**Statistical analysis.** For animal studies, sample size for experiments were determined empirically based on previous studies to ensure appropriate statistical power. Animals for each group of experiments were chosen randomly. No animals were excluded from statistical analysis, and the investigators were not blinded in the studies. The statistical analysis of different groups is realized using the Student's unpaired $t$-test or one-way analysis of variance (ANOVA) followed by Tukey post hoc test or two-way ANOVA. SPSS software (IBM) was used for all analyses.

**Data availability.** mRNA-sequencing data that support the findings of this study have been deposited in Gene Expression Omnibus (GEO) with the primary accession codes GSE83338.

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

## Acknowledgements

We thank members of the Chromatin Dynamics Research Center for technical assistance and discussions, J.B.K. for providing PPARγ constructs. We thank the NCC Animal Sciences Branch for excellent guidance and assistance with the performed mouse experiments. This work was supported by Creative Research Initiatives Program (Research Center for Chromatin Dynamics, 2009-0081563) to S.H.B.; Global PH.D Fellowship Program (NRF-2011-0008101) to K.K. and (NRF-2012H1A2A1009905) to Y.S.Y.; Korea Mouse Phenotyping Project (2013M3A9D5072550) to S.G.Y., I.Y.K., J.K.S. and S.F.; Basic Science Research Program (NRF-2015R1D1A1A01058037) to K.B., (NRF-2014R1A6A3A04057910) to H.K.; and (NRF -2015R1C1A1A01052195) to S.F.; Medical Research Center (NRF-2014R1A5A2010008) to S.-S.I.; Institute for Basic Science (IBS-R013-G1) to D.H. from the National Research Foundation (NRF) grant funded by the Korea government (MSIP). H.L. was supported by the National Cancer Center Grant (NCC-1310100).

## Author contributions

K.K., K.I.K., H.L., S.F. and S.H.B. designed the experiments. K.K., K.B., S.K.O. Y.J. and H.L. generated and maintained RORα$^{f/f}$ and RORα$^{LKO}$ mice. S.G.Y., I.Y.K. and J.K.S. performed mouse metabolic cage assay. J.-S.L. and S.-S.I. provided *Ppara*-null primary hepatocyte. K.K., S.K.O. and S.F. analyzed mouse phenotype. K.K. performed histological assays. K.K., K.B., S.K.O. and H.K. performed the cell biology and biochemistry experiments. K.K., Y.S.Y., J.B. and D.H. performed mRNA-sequencing preparation and analysis. K.K., Y.S.Y., D.H., S.F. and S.H.B. wrote the manuscript. All authors contributed to data analysis.

## Additional information

**Competing interests:** The authors declare no competing financial interests.

