## [Peer Review file · Nature Communications]

Reviewers' comments:

Reviewer #1 (Remarks to the Author):

In this manuscript Kim et al describe the phenotype of liver-specific RORa KO mice upon HFD challenge. They found that RORa negatively affects the PPARg signaling pathway. In the absence of RORa, PPARg signaling is dysregulated, leading to hepatic steatosis, insulin resistance and obesity upon HFD, suggesting that targeting RORa might be beneficial to treat metabolic disorders.

Overall the topic is timely and interesting. The data presented are novel and the phenotype is a profound one, however there is a major conceptual concern remains and several major issues concerning the mechanistic part, which would need further experimentation. Although the data presented is credible, the link between the in vivo phenotype and the mechanistic explanation is weak and includes multiple leaps of faith.

The conceptual concern is that the authors insufficiently explore the contribution of other PPARs present in the liver and might contribute to the phenotype. The main PPAR isoform present in hepatocytes is PPARa. Its expression levels is much higher than that of PPARg in hepatocytes and its role is the regulation of lipid oxidation and participates in the fasting response. This cannot be ignored and must be addressed experimentally using the available PPARa KO animals. The fact that the authors show that PPARa is not bound to two enhancers they examine on Figure 5a does not exclude the possibility that PPARa signaling and target genes are also affected. The experiment on Figure 5a does not have a positive control for PPARa binding, so it is not clear how efficiently the antibody works. As the authors now acknowledge PPARa and g can bind to similar binding sites, so the contribution of RORa to PPARa signaling also must be addressed. However what they show does not go far enough to address this issue. There is no reason to believe that the PPARa binding would not be impacted and it would be a PPARg-selective effect.

Major issues:

1. PGC1a is not a "prototypical" PPARg co-factor, in fact its major biological effect is via ERRs. Therefore it would be important to assess other co-factors such as p160 family members.
2. On figure 5B and C, it would be important to show at least one negative control region.
3. What is the relevance of the RORa+Release ChIP experiments? This is not discussed in sufficient detail.
4. According to the wash out experiments, it seems that RORa binding might provide some type of epigenetic memory on PPARg targeted gene promoters. How would a second stimulation of RSG affect the binding of PPARg and PGC1a? Is there a blunted response to RSG upon second stimulation at the gene expression level?
5. It would be important to indicate the exact genomic locations for the ChIP experiments.
6. RORa seems to be recruited to the PPARE in a RSG-dependent manner (Figure 5B). Does the presence of PPARg is required for RORa recruitment and repression of PPARg targets?
7. Is it possible that PPARg and RORa interact with each other? The authors should consider doing reChIP to clarify that PPARg and RORa can bind to the same genomic region in one complex.

Reviewer #2 (Remarks to the Author):

In this manuscript, Kim and colleagues study the effect of hepatic deletion of RORa. In particular the authors focused on metabolic alterations occurring in response to HFD. RORa LKO mice in fact showed increased hepatic steatosis compared to floxed mice, paralleled by an impairment of insulin sensitivity. Transcriptome analysis showed that PPAR signaling pathway is affected by hepatic deletion of RORa. In fact the expression of several genes involved in gluconeogenesis was upregulated in KO mice. To explain this phenotype, the authors proposed that RORa interacts with HDAC3 and this complex blocks PPARγ-mediated transcriptional activation, because it competes with PPARγ/PGC1α for the binding to PPARE regions in PPARγ target genes promoter. Treatment of

ROR α LKO mice fed HFD with PPAR γ antagonist reduced body weight gain and liver steatosis, thus recovering the phenotype.

The topic of this manuscript is potentially interesting, however the authors should elucidate further the mechanistic insights of the story. Substantial additional work is needed to test whether the proposed mechanism is correct.

Major comments:

1) The first result showed by the authors is the different behavior with CD or HFD. They showed differences in body weight gain in floxed and KO mice with the two different diets, but they did not show any data about liver morphology in mice fed CD. Considering that their model shows some analogies with the Hdac3-liver KO published by the group of Lazar (Nat. Med., 2012) in which it has been demonstrated that hepatic ablation itself is able to increase steatosis, it would be appropriate to show also H&E and ORO staining and gene expression analysis (gluconeogenesis, lipogenesis and lipid sequestration genes) in livers from floxed and KO fed CD.

2) In the same paper the Lazar's group demonstrated that upon Hdac3 ablation perilipin gene is upregulated in liver, contributing to lipid sequestration and thus to amelioration of glucose tolerance. So why the increased expression of perilipin in ROR α LKO mice did not improve glucose tolerance? The authors should comment their results on perilipin in the discussion in the light of previous observations by Lazar's group.

3) In figure 3 the authors showed the result of RNAseq experiment. They should also include a heat map in the manuscript showing the 4 different groups they analyzed and not simply the heat map of the comparisons (Fig. 3a). This will help elucidate the different global profile of gene expression upon different nutritional conditions. In figure 3d in fact they reported gene expression of several genes, and it is surprising to notice that expression of all these important genes was not affected by high fat feeding. Do the authors have any explanation? Furthermore the authors considered this set of genes as known Ppar targets, and in the following figures (fig. 5) they focused only on Cd36 and Cpt1b. However, among genes upregulated upon ROR α ablation there are other more interesting targets. One of them is perilipin, whose important role in the establishment of Hdac3 KO mice has been demonstrated. So, considering that Plin gene is target of Ppar, why did the authors not focus on this gene? Another point is why the authors pointed the attention on Cpt1b, which is typically not expressed in the liver (hepatic isoform is Cpt1a), and it is known to be target of PPAR γ ? The authors should explain carefully all these issues in the manuscript. Moreover, activation of Cpt1b gene (that is part of fatty acid beta-oxidation pathway) seems to be inconsistent with the lipid accumulation observed in livers of ROR α KO mice.

4) In figure 5 authors analyzed recruitment of different nuclear receptors/transcription factors on PPRE in Cd36 and Cpt1b promoter. They performed experiments in cell cultures. It would be appropriate to show also gene expression profile of these mRNAs in response to different experimental conditions (Rosi treatment, ROR α or PPAR γ knock-down), to verify whether gene expression profile paralleled ChIP results. At this regard a ChIP analysis of PolII on these promoters would be informative. The ChIP analysis showed in fig 5c should be performed also on Cd36 PPRE region, because this information would be very relevant to characterize the phenotype of the mouse model. Moreover, considering the key role played by Plin in Hdac3 KO mice, and considering that it has been demonstrated that ROR α inhibits activation of the perilipin promoter by PPAR γ (The Orphan Nuclear Receptor ROR α Restrains Adipocyte Differentiation through a Reduction of C/EBP β Activity and Perilipin Gene Expression. Ohoka et al., DOI: <http://dx.doi.org/10.1210/me.2008-0277>) the authors must look at PPRE region in Plin genes (all ChIP analyses must also be performed on this promoter).

5) The authors assert that ablation of ROR α allow PPAR γ recruitment on target genes, determining establishing of fatty liver phenotype. However, it has been demonstrated that PPAR γ activation by thiazolidinediones can ameliorate hepatic steatosis and insulin resistance, and that it lowers triglycerides content (Sci Rep. 2016 Aug 22;6:31542. doi: 10.1038/srep31542. Reduction of obesity-associated white adipose tissue inflammation by rosiglitazone is associated with reduced non-alcoholic fatty liver disease in LDLr-deficient mice. Mulder et al.). How do the authors explain this discrepancy? They should comment this aspect in the discussion.

- 6) ROR α LKO phenotype pops up when mice are HFD, since when mice are fed CD they showed no differences from floxed mice. It is fundamental to investigate whether the same molecular events (higher recruitment of PPAR γ /PGC1 α and lower recruitment of HDAC3) also occur in mice fed CD. It is possible that in CD mice the phenotype is not induced because of the low availability of fatty acids as PPAR γ ligands under this dietary condition. Therefore, the authors should perform ChIP analysis in both CD and HFD mice to address this important point.
- 7) What happens to genes of de novo fatty acid synthesis (Chrebp, Srebp1, Fasn, Acaca etc.) in the ROR α KO mouse model?
- 8) All statistical analyses should be revised. It is totally missing in figure 3. Comparisons among three or more groups require 1way or 2way ANOVA.
- 9) In figure 5d the authors showed higher recruitment of PPAR γ on Cd36 and CPT1b PPRE in siPPAR γ +vehicle treated cells, compared to siNS+vehicle treated cells. I would expect no signal at all in cells in which they knocked down PPAR γ , or at least a lower signal compared to siNS treated cells.

Minor comments:

- 1) It is not appropriate to refer to PPAR γ as an orphan receptor, since several ligands have been identified and described in different publications.
- 2) The authors should indicate the experimental paradigm they used in vivo studies (which type of diet and how long was diet challenge).
- 3) In figure1 authors showed that hepatic ablation of ROR α increased inflammatory genes in eWAT and reduced expression of thermogenic genes in BAT. Is this barely the result of the increased body weight (and reduced insulin sensitivity) or could it be a consequence of a loss of functional liver-adipose axis? The authors should comment these results in the discussion.
- 4) In figure 4f and 4g, the first two bars are referring to the same experimental conditions. Therefore, why is the fold-increase in R.L.U. induced by PPAR γ /PGC1 α so different in the two experiments (50 fold in panel f and less than 2 fold in panel g)?
- 5) In figure 5c IgG or GFP (negative control Ab) condition is missing.
- 6) Check primers list (some primers sequences are missing, e.g. Primers for Cpt1b mRNA expression).

Reviewer #3 (Remarks to the Author):

Kim et al developed and used hepatocyte-specific ROR α deficient mice, RNA seq, and ChIP-Seq to compile a narrative indicating a primary inhibitory role for ROR α in liver triglyceride storage and liver injury through an HDAC3-dependent mechanism that inhibits PPAR γ signaling.

The data are reasonably compelling, but like many manuscripts characterizing knockout models that have adiposity phenotypes, this study lacks differentiation of chicken from egg.

1. This is to say, does hepatocyte-specific deletion of ROR α increase caloric intake and/or decrease energy expenditure, causing obesity, insulin resistance, and fatty liver? Or are the observed phenotypes entirely attributable to effects of ROR α on PPAR γ signaling in a manner that regulates lipid metabolism in hepatocytes? Given the high fat diet induced obesity phenotype, the latter seems unlikely, and the authors do nothing to address this rather gaping hole.

How does hepatocyte-selective deletion of a transcription factor cause such significant reprogramming of adipose tissue and systemic energy homeostasis? This needs to be substantively addressed. The Discussion on sg mice is not particularly helpful, because as the authors recognize, this model lacks ROR α in all tissues including those of the nervous system.

In addition:

2. The experiment presented in Fig. 4e, Co-immunoprecipitation with ROR α and HDAC3, should

also be performed examining endogenous proteins, not only those that are over-expressed. Clearly RORa, HDAC3, and PPARg signaling are connected on several promoters, and the experiments presented in Fig. 4f and Fig. 5 help demonstrate this, but the data do not definitively support the competition model in Fig. 6h. Experiments to address would include HDAC3 knockdown and/or determination of HDAC3 recruitment in RORa knockout liver.

3. In Fig. 6, why does GW9662 decrease the floor on liver TAG and FAO gene regulation (6e-f), but not lipogenesis gene regulation (6g)? Specifically, there appears to be an RORa independent component of the GW effect on the genes studied in 6f, and on liver TAG. This question addresses the greater concern that the authors may oversimplify the molecular mechanism among transcription factors studied.

Point-by-point response to the Reviewers' comments:

We thank the Reviewers for their positive comments and for identifying some standing issues. Here we are submitting a revised version of our manuscript based on their suggestions. Our detailed response to each of the Reviewer's points is reported below.

#1 Reviewer's comments:

In this manuscript Kim et al describe the phenotype of liver-specific ROR α KO mice upon HFD challenge. They found that ROR α negatively affects the PPAR γ signaling pathway. In the absence of ROR α , PPAR γ signaling is dysregulated, leading to hepatic steatosis, insulin resistance and obesity upon HFD, suggesting that targeting ROR α might be beneficial to treat metabolic disorders.

Overall the topic is timely and interesting. The data presented are novel and the phenotype is a profound one, however there is a major conceptual concern remains and several major issues concerning the mechanistic part, which would need further experimentation. Although the data presented is credible, the link between the in vivo phenotype and the mechanistic explanation is weak and includes multiple leaps of faith.

The conceptual concern is that the authors insufficiently explore the contribution of other PPARs present in the liver and might contribute to the phenotype. The main PPAR isoform present in hepatocytes is PPAR α . Its expression levels is much higher than that of PPAR γ in hepatocytes and its role is the regulation of lipid oxidation and participates in the fasting response. This cannot be ignored and must be addressed experimentally using the available PPAR α KO animals. The fact that the authors show that PPAR α is not bound to two enhancers they examine on Figure 5a does not exclude the possibility that PPAR α signaling and target genes are also affected. The experiment on Figure 5a does not have a positive control for PPAR α binding, so it is not clear how efficiently the antibody works. As the authors now acknowledge PPAR α and γ can bind to similar binding sites, so the contribution of ROR α to PPAR α signaling also must be addressed. However what they show does not go far enough to address this issue. There is no reason to believe that the PPAR α binding would not be impacted and it would be a PPAR γ -selective effect.

We thank the Reviewer for highlighting this issue and for his/her suggestion. We have evaluated the potential roles of ROR α on PPAR α signaling network.

We first tested classical PPAR α target gene expressions, including *Acox1* and *Fgf21* in the HFD-fed ROR α^{ff} and ROR α^{LKO} mouse model. We did not find robust significant differences in the expression of *Acox1* and *Fgf21* among the analyzed genotypes (revised supplementary Fig. 2a). Given that PPAR α signaling is largely activated in the setting of energy deprivation, we next tested PPAR α signaling in fasted/refed ROR α^{ff} and ROR α^{LKO} mice. Recently, Dr. David Moore's lab has reported a beautiful paper (Lee J.M. *et al. Nature* 2014) that PPAR α is a nutrient sensing nuclear receptor to coordinate

autophagic gene expressions. Based on the experimental setting from the Moore's report, we tested the hepatic expression profiles of *Acox1* and *Fgf21* in $ROR\alpha^{fl/fl}$ and $ROR\alpha^{LKO}$ mice first. We observed little or no difference of gene expression profile between genotypes in fasting status, indicating that PPAR α activation was not further enhanced in the absence of ROR α (revised supplementary Fig. 2b). We performed ChIP assay to determine the recruitment of PPAR α to the PPRE of those genes. The ChIP data revealed no significant differences in PPAR α recruitment among the genotypes (revised supplementary Fig. 2c), suggesting that PPAR α recruitment to the PPRE region of *Acox1* and *Fgf21* genes was not affected by the presence or absence of ROR α .

Besides *Acox1* and *Fgf21*, Dr. Moore's paper has reported that PPAR α is able to coordinate several key autophagic genes, including *LC3a* and *Sesn2* (Lee J.M. *et al. Nature* 2014). Consistently, expression of *LC3a* and *Sesn2* was similar among the genotypes (revised supplementary Fig. 2d). Thus, we did not observe notable activation of PPAR α in the fasted or HFD-fed $ROR\alpha^{LKO}$ mice. Thus, we believe that ROR α mainly controls PPAR γ transcriptional network to maintain hepatic homeostasis in response to HFD.

The Reviewer suggested studying *Ppar α* -null mice for the revised manuscript. It is absolutely important and reasonable question to delineate the pathogenesis of fatty liver in response to HFD. However, *Ppar α* -null mice have been reported to impair fatty acid oxidation systems in the liver and thus develop remarkably hepatic steatosis (Hashimoto T. *et al. J. Biol. Chem.* 2000; Kersten S. *et al. J. Clin. Invest.* 1999). If ROR α suppressed PPAR α signaling pathway as it suppresses PPAR γ signaling pathway, hepatic steatosis phenotype of $ROR\alpha^{LKO}$ mice would not be observed. In our mouse model, PPAR α signaling has not been either further increased or decreased in HFD-fed or fasted $ROR\alpha^{LKO}$ mice, indicating that ROR α would not play a critical role to mediate hepatic PPAR α signaling pathway.

Moreover, the expression of *Cd36* and *Plin2* genes was similar among the genotypes when FFA was treated to activate the PPAR γ signal in the primary hepatocyte from WT and *Ppar α* -null mice (revised supplementary Fig. 7c). In other words, it is now confirmed that PPAR α in the liver does not have influence on PPAR γ transcriptional network in the setting of HFD. Taken together, we propose that ROR α mainly controls PPAR γ transcriptional network rather than PPAR α signaling in response to HFD, and PPAR γ transcriptional network is not affected by PPAR α . Thus, we would like to mention that ROR α regulates PPAR γ transcriptional network independent of PPAR α .

Nevertheless, we completely agree with the Reviewer on the importance of studying the roles of ROR α to modulate PPAR α transcriptional network. We would like to mention that ROR α mainly coordinates PPAR γ transcriptional network in response to HFD, and PPAR γ signaling has been largely enhanced in the absence of hepatic ROR α . Altogether, we have proposed that though PPAR α is a major PPAR isoform in the liver, ROR α has a repressive role to attenuate PPAR γ transcriptional network in response to environmental stress, such as HFD to prevent against diet-induced hepatic steatosis and obesity.

Major issues:

1. PGC1 α is not a “prototypical” PPAR γ co-factor, in fact its major biological effect is via ERRs. Therefore it would be important to assess other co-factors such as p160 family members.

We thank to the Reviewer. We performed reporter assay using p160 family members. The new data have been incorporated in revised Fig. 4b, Fig. 4h, supplementary Fig. 3a, and supplementary Fig. 3e. Our data have revealed that ROR α is able to repress p160 family-mediated PPAR γ activation.

2. On figure 5B and C, it would be important to show at least one negative control region.

We thank to the Reviewer. We performed ChIP assay to show the negative control. The data have been incorporated in revised Fig. 5b.

3. What is the relevance of the Rosi.+ Release ChIP experiments? This is not discussed in sufficient detail.

We thank the Reviewer for other great point on the relevance of the Rosi.+Release ChIP experiments. The relevance of Rosi.+washout (Release) ChIP experiments was originally designed to determine if ROR α has repressive roles to mediate PPAR γ activation in natural context. Instead of PPAR γ antagonist treatment, we designed the experimental setting to repress Rosi.-mediated PPAR γ activation naturally. Thus, we just washed out the rosiglitazone, waited until Rosi.-effect has been diminished, and performed ChIP assay whether ROR α recruitment was affected after effect of rosiglitazone has been diminished. We added our experimental scheme of Rosiglitazone washout in revised supplementary Fig. 5a. Also, new data in the setting of Rosiglitazone washout have been incorporated in revised Fig. 5, Fig. 6a and supplementary Fig. 5.

To be convinced with washout results, we also tested repressive roles of ROR α in the presence of GW9662, a well-known synthetic PPAR γ antagonist. Similar to our Rosi.+washout ChIP data, the recruitment of ROR α were dramatically increased, whereas PPAR γ recruitment was largely diminished by GW9662 treatment to the promoters of several PPAR γ target genes. These data have been incorporated in revised supplementary Fig. 6.

4. According to the wash out experiments, it seems that ROR α binding might provide some type of epigenetic memory on PPAR γ targeted gene promoters. How would a second stimulation of RSG affect the binding of PPAR γ and PGC1 α ? Is there a blunted response to RSG upon second stimulation at the gene expression level?

We thank the Reviewer for great point on our manuscript. Changes in recruitment of ROR α and PPAR γ by second stimulation of rosiglitazone may explain the repression mechanism of PPAR γ signaling by ROR α . We performed ChIP assay after second stimulation of rosiglitazone. As a result, PPAR γ was recruited to target gene promoter again and recruitment of ROR α was largely decreased. We also checked the gene

expression level after second stimulation of rosiglitazone. Repressed PPAR γ target gene expressions were restored to or increased than first stimulation of rosiglitazone treatment (Figure to the Reviewer). These results suggest that ROR α interferes with the recruitment of PPAR γ on the target gene promoters to inhibit PPAR γ activation rather than regulates PPAR γ signaling through modulation of epigenetic memory mediated by ROR α .

Figure to the Reviewer

(a) Expression levels of PPAR γ target genes in the absence or presence of ROR α in Hep3B

cells with or without Rosiglitazone treatment for 24 hr and washout 8 hr and second rosiglitazone stimulation for 24hr. (b) ChIP assays were performed in the absence or presence of ROR α on *SCD* and *PLIN2* promoters in Hep3B cells with or without Rosiglitazone treatment for 24 hr and washout 8 hr and second rosiglitazone stimulation for 24hr.

5. It would be important to indicate the exact genomic locations for the ChIP experiments.

We thank the Reviewer. We incorporated genomic locations for the ChIP experiments in revised Fig. 5a and Fig. 5b.

6. ROR α seems to be recruited to the PPARE in a RSG-dependent manner (Figure 5B). Does the presence of PPAR γ is required for ROR α recruitment and repression of PPAR γ targets?

We thank the Reviewer. We performed ChIP assay using PPAR γ siRNA. According to our updated data, the recruitment of ROR α and HDAC3 was not affected by the presence of PPAR γ in basal condition. In the condition of PPAR γ activation, PPAR γ was largely recruited, whereas ROR α /HDAC3 recruitment was not affected. However, we clearly noticed that reduction of PPAR γ recruitment by PPAR γ siRNA resulted in the increase of ROR α and HDAC3 recruitment to the promoters of PPAR γ target genes. These data indicate that the presence of PPAR γ is not required for ROR α recruitment. These data have been incorporated in the revised Fig. 6b and supplementary Fig. 7d.

7. Is it possible that PPAR γ and ROR α interact with each other? The authors should consider doing reChIP to clarify that PPAR γ and ROR α can bind to the same genomic region in one complex.

We thank the Reviewer. PPAR γ and ROR α do not interact with each other. The new data has been updated in revised supplementary Fig. 3c. We clearly showed that ROR α does not bind to PPAR γ , suggesting that ROR α may compete with PPAR γ to be recruited to PPARE region. To demonstrate this issue, we performed ChIP assay using synthetic WT PPARE promoter and PPARE deleted mutant promoter. The recruitment of both ROR α and PPAR γ were largely diminished in PPARE deleted mutant promoter region, indicating that ROR α may be able to bind to PPARE by itself. These data have been incorporated in revised supplementary Fig. 3b.

We thank the Reviewer for suggesting the ReChIP assay, which turned out to be great experiment. We agreed with the Reviewer's point and we performed ReChIP assay to clarify if PPAR γ and ROR α are able to bind to the same genomic region and their recruitments to promoter of target genes are mutually exclusive. Our data clearly showed that ROR α and PPAR γ were not recruited to the same genomic region in one complex. These data have been incorporated in revised supplementary Fig. 7e.

We thank the Reviewer for the very helpful comments, which have clearly enhanced the rigor of further supporting our hypothesis. We hope you like the revised manuscript.

Reviewer #2 (Remarks to the Author):

In this manuscript, Kim and colleagues study the effect of hepatic deletion of ROR α . In particular the authors focused on metabolic alterations occurring in response to HFD. ROR α LKO mice in fact showed increased hepatic steatosis compared to floxed mice, paralleled by an impairment of insulin sensitivity. Transcriptome analysis showed that PPAR signaling pathway is affected by hepatic deletion of ROR α . In fact the expression of several genes involved in gluconeogenesis was upregulated in KO mice. To explain this phenotype, the authors proposed that ROR α interacts with HDAC3 and this complex blocks PPAR γ -mediated transcriptional activation, because it competes with PPAR γ /PGC1 α for the binding to PPRE regions in PPAR γ target genes promoter. Treatment of ROR α LKO mice fed HFD with PPAR γ antagonist reduced body weight gain and liver steatosis, thus recovering the phenotype.

The topic of this manuscript is potentially interesting, however the authors should elucidate further the mechanistic insights of the story. Substantial additional work is needed to test whether the proposed mechanism is correct.

Major comments:

1) The first result showed by the authors is the different behavior with CD or HFD. They showed differences in body weight gain in floxed and KO mice with the two different diets, but they did not show any data about liver morphology in mice fed CD. Considering that their model shows some analogies with the Hdac3-liver KO published by the group of Lazar (Nat. Med., 2012) in which it has been demonstrated that hepatic ablation itself is able to increase steatosis, it would be appropriate to show also H&E and ORO staining and gene expression analysis (gluconeogenesis, lipogenesis and lipid sequestration genes) in livers from floxed and KO fed CD.

We thank the Reviewer for his/her comments. The difference of metabolic status between ROR α ^{f/f} and ROR α ^{LKO} mice fed CD was not observed. There was no significant difference in liver histology and energy expenditure. Gene expression analysis showed little or no significant difference between WT and KO mice fed CD. We incorporated all new data in revised supplementary Fig. 1.

2) In the same paper the Lazar's group demonstrated that upon Hdac3 ablation perilipin gene is upregulated in liver, contributing to lipid sequestration and thus to amelioration of glucose tolerance. So why the increased expression of perilipin in ROR α LKO mice did not improve glucose tolerance? The authors should comment their results on perilipin in the discussion in the light of previous observations by Lazar's group.

We thank the Reviewer for the great point on our manuscript. According to Lazar's group, Perilipin 2 was largely upregulated. We observed that the expression of Perilipin, especially Perilipin 2 was largely upregulated in the liver. These data have been incorporated in revised Fig. 2f.

We agree with the Reviewer that HDAC3 plays a key role to regulate hepatic glucose/lipid homeostasis as Lazar group has reported. However, a previous report has

reported that liver-specific PPAR γ deletion led to improved glucose tolerance (Morán-Salvador E. *et al. FASEB J.* 2011). Our mouse model has impaired ROR α -mediated HDAC3 transcriptional repression as well as ROR α itself, leading to interfering with hepatic homeostasis of PPAR γ transcriptional network. With HFD, PPAR γ signal was largely upregulated in HFD-fed ROR α^{LKO} mice. Therefore, disturbed regulatory mechanism of PPAR γ signaling in HFD-fed hepatic ROR α KO would be the main cause of the insulin resistance and glucose intolerance. We discussed this issue in the section of Discussion.

Then, ablation of *Plin2* in *Lep^{ob/ob}* mice also has been exhibited to improve glucose tolerance (Chang B. H-J. *et al. J. Lipid Res.* 2010). Therefore, Perilipin 2 regulates glucose tolerance depending on the context. We hope that our study will provide further information in understanding the role of Perilipin 2 in the liver.

3) In figure 3 the authors showed the result of RNAseq experiment. They should also include a heat map in the manuscript showing the 4 different groups they analyzed and not simply the heat map of the comparisons (Fig. 3a). This will help elucidate the different global profile of gene expression upon different nutritional conditions.

We thank the Reviewer for his/her critical comments. When heatmap is drawn with four groups (ROR $\alpha^{\text{fl/fl}}$ _CD/ROR $\alpha^{\text{fl/fl}}$ _HFD/ROR α^{LKO} _CD/ ROR α^{LKO} _HFD), it is intuitive to look at the gene expression patterns according to nutritional condition as the Reviewer points out. However, we focused on the fact that HFD-fed ROR α^{LKO} mice exhibited more obese phenotype than the HFD-fed ROR $\alpha^{\text{fl/fl}}$ mice. Because we did not observe phenotypical difference among genotypes fed CD, we wanted to find gene lists that have been remarkably changed among genotype fed HFD rather than fed CD,. We hypothesize that remarkable change of gene expression only in HFD-fed ROR α^{LKO} mice would contribute to hepatic steatosis and obese phenotype. Thus we designed analysis of RNA-seq to compare the difference of gene expression changes among genotypes fed HFD with those among genotypes fed CD. While gene expression profiles were similar among genotype fed CD, group1 especially exhibited significant increase of gene expression levels only in HFD-fed ROR α^{LKO} mice. Therefore, we have drawn heatmap using the comparison method for the purpose of easy understanding.

In figure 3d in fact they reported gene expression of several genes, and it is surprising to notice that expression of all these important genes was not affected by high fat feeding. Do the authors have any explanation?

We thank the Reviewer for his/her critical comments. The expression levels of *Cd36* and *Elovl3* were significantly increased in HFD-fed ROR $\alpha^{\text{fl/fl}}$ mice. Therefore, the significance between CD- and HFD-fed ROR $\alpha^{\text{fl/fl}}$ mice was indicated in the figure. In addition, the expression levels of *Slc27a1* and *Pdk4* were increased in mouse samples of different cohort. The expression level of *Acot2* was also affected by HFD. We modified the main figure and text accordingly in revised manuscript.

Furthermore the authors considered this set of genes as known Ppar targets, and in the following figures (fig. 5) they focused only on *Cd36* and *Cpt1b*. However, among genes

upregulated upon ROR α ablation there are other more interesting targets. One of them is perilipin, whose important role in the establishment of Hdac3 KO mice has been demonstrated. So, considering that Plin gene is target of Ppar, why did the authors not focus on this gene?

Another point is why the authors pointed the attention on Cpt1b, which is typically not expressed in the liver (hepatic isoform is Cpt1a), and it is known to be target of PPAR γ ? The authors should explain carefully all these issues in the manuscript. Moreover, activation of Cpt1b gene (that is part of fatty acid beta-oxidation pathway) seems to be inconsistent with the lipid accumulation observed in livers of ROR α KO mice.

We performed numerous mechanism studies using *Plin2* promoter region. All updated data have been incorporated in revised supplementary Fig. 4a, 4b, 5b, 5c, 6, and 7.

The Reviewer's point is great that activation of *Cpt1b* is inconsistent with the lipid accumulation in the liver. We believe that this is the main point of the paradox of PPAR γ activation. Though PPAR γ activation has shown to reduce blood glucose level and hepatic gluconeogenesis, but improve glucose tolerance (Saltiel A.R. & Olefsky J.M. *Diabetes* 1996; Way J.M. *et al. Endocrinology* 2001; Festuccia W.T. *et al. Am. J. Physiol. Endocrinol. Metab.* 2014), several papers have reported that PPAR γ activation leads to hepatic steatosis (Schadinger S.E. *et al. Am. J. Physiol. Endocrinol. Metab.* 2005; Mora'n-Salvador E. *et al. FASEB J.* 2011; Lee Y.J. *et al. PNAS* 2012). Thus, the physiological impacts of PPAR γ activation are not simple to understand. We here studied the functional roles of ROR α to delineate hepatic PPAR γ -mediated transcriptional network. We hope that our study will help to understand the paradoxical roles of PPAR γ in the system.

As Reviewer mentioned here, *Cpt1a* is the major isoform of *Cpt1* in the liver while *Cpt1b* is the specific isoform in the skeletal muscle. Quite interestingly, the expression of *Cpt1b* has been observed to be remarkably upregulated in HFD-fed ROR α ^{LKO} mice by RNA seq and qRT-PCR. We still do not know the exact mechanism of how *Cpt1b* expression level has been upregulated in HFD-fed ROR α ^{LKO} mice. It has been known that *Cpt1b* plays a key role to inhibit progression of fatty liver via beta-oxidation pathway as noted by the Reviewer. Though we hypothesized that *Cpt1b* upregulation would be resulted from uncontrolled PPAR γ activation in the liver, we absolutely agree with the Reviewer's point, the role of *Cpt1b* is not well appropriate for the scope of our hypothesis. Therefore, we replaced all *Cpt1b* data with *Plin2* or *Scd1* data.

4) In figure 5 authors analyzed recruitment of different nuclear receptors/transcription factors on PPRE in Cd36 and Cpt1b promoter. They performed experiments in cell cultures. It would be appropriate to show also gene expression profile of these mRNAs in response to different experimental conditions (Rosi treatment, ROR α or PPAR γ knock-down), to verify whether gene expression profile paralleled ChIP results.

We added updated data in revised supplementary Fig. 5b, Fig. 6a, and Fig. 7a.

At this regard a ChIP analysis of PolII on these promoters would be informative. The ChIP

analysis showed in fig 5c should be performed also on Cd36 PPRE region, because this information would be very relevant to characterize the phenotype of the mouse model. Moreover, considering the key role played by Plin in Hdac3 KO mice, and considering that it has been demonstrated that ROR α inhibits activation of the perilipin promoter by PPAR γ (The Orphan Nuclear Receptor ROR α Restrains Adipocyte Differentiation through a Reduction of C/EBP β Activity and Perilipin Gene Expression. Ohoka et al., DOI: <http://dx.doi.org/10.1210/me.2008-0277>) the authors must look at PPRE region in Plin genes (all ChIP analyses must also be performed on this promoter).

We have incorporated new data in revised Fig. 5, Fig. 6, supplementary Fig. 4a, supplementary Fig. 4b, supplementary Fig. 5c, supplementary Fig. 6b and supplementary Fig.7b.

5) The authors assert that ablation of ROR α allow PPAR γ recruitment on target genes, determining establishing of fatty liver phenotype. However, it has been demonstrated that PPAR γ activation by thiazolidinediones can ameliorate hepatic steatosis and insulin resistance, and that it lowers triglycerides content (Sci Rep. 2016 Aug 22;6:31542. doi: 10.1038/srep31542. Reduction of obesity-associated white adipose tissue inflammation by rosiglitazone is associated with reduced non-alcoholic fatty liver disease in LDLr-deficient mice. Mulder et al.). How do the authors explain this discrepancy? They should comment this aspect in the discussion.

We really thank the Reviewer for his/her comments. According to the above mentioned papers, rosiglitazone treatment reduces hepatic steatosis and ameliorates insulin resistance in HFD-fed LDLr-deficient mice. However, thiazolidinediones-induced body weight gain is a well-known side effect of PPAR γ activation in human patients as well as mouse models (Lehrke M. & Lazar M.A. *et al. Cell* 2005; Gerstein H.C. *et al. Lancet* 2006; Kahn S.E. *et al. N. Engl. J. Med.* 2006)

In general, the expression of PPAR γ is very low in human and mouse liver. Interestingly, the expression level of hepatic PPAR γ 2 is significantly upregulated in obese rodent model (Vidal-Puig A. *et al. J. Clin. Invest.* 1996), suggesting that PPAR γ plays a key role to develop hepatic steatosis. Accordingly, inhibition of PPAR γ signaling and hepatic deficiency of PPAR γ in ob/ob mice and have shown to ameliorate fatty liver (Yamauchi T. *et al. J. Clin. Invest.* 2001; Matsusue K. *et al. J. Clin. Invest.* 2003). Thus, these results strongly suggested that the PPAR γ signaling pathway is involved in diet-induced hepatic steatosis, and hepatic lipid accumulation would be prevented by suppression of PPAR γ transcriptional network in the liver.

While PPAR γ in adipose tissue has been well established to induce adipogenesis and regulate fatty acid metabolism to improve glucose homeostasis, the molecular mechanism of how PPAR γ induces hepatic steatosis still remains unclear. Thus, our novel findings that ROR α mediates PPAR γ transcriptional network to maintain hepatic glucose/lipid metabolism to prevent against diet-induced obesity are pretty important. We have discussed this issue in the section of discussion.

6) ROR α LKO phenotype pops up when mice are HFD, since when mice are fed CD they

showed no differences from floxed mice. It is fundamental to investigate whether the same molecular events (higher recruitment of PPAR γ /PGC1 α and lower recruitment of HDAC3) also occur in mice fed CD. It is possible that in CD mice the phenotype is not induced because of the low availability of fatty acids as PPAR γ ligands under this dietary condition. Therefore, the authors should perform ChIP analysis in both CD and HFD mice to address this important point.

We added ChIP data from CD-fed mice in revised supplementary Fig. 4.

7) What happens to genes of de novo fatty acid synthesis (Chrebp, Srebp1, Fasn, Acaca etc.) in the ROR α KO mouse model?

We analyzed those gene expression profiles and added them in revised Fig. 2f.

8) All statistical analyses should be revised. It is totally missing in figure 3. Comparisons among three or more groups require 1way or 2way ANOVA.

We performed statistical analysis using 1way or 2way ANOVA and described in each figure legend.

9) In figure 5d the authors showed higher recruitment of PPAR γ on Cd36 and CPT1b PPRE in siPPAR γ +vehicle treated cells, compared to siNS+vehicle treated cells. I would expect no signal at all in cells in which they knocked down PPAR γ , or at least a lower signal compared to siNS treated cells.

We thank the Reviewer and the Reviewer's suggestion is absolutely right. The qRT-PCR value was low for the recruitment of PPAR γ , a slight variation in the qRT-PCR value led to a big S.E.M. in the figure. As the Reviewer suggested, we carefully analyzed the data again, and lower signal was observed. New improved data have been incorporated in revised Fig. 6b.

Minor comments:

1) It is not appropriate to refer to PPAR γ as an orphan receptor, since several ligands have been identified and described in different publications.

We changed the terminology.

2) The authors should indicate the experimental paradigm they used in vivo studies (which type of diet and how long was diet challenge).

We added experimental paradigm in the Method.

3) In figure1 authors showed that hepatic ablation of ROR α increased inflammatory genes in eWAT and reduced expression of thermogenic genes in BAT. Is this barely the result of the increased body weight (and reduced insulin sensitivity) or could it be a consequence of a loss

of functional liver-adipose axis? The authors should comment these results in the discussion.

Dr. Auwerx group has previously reported that bile acid signaling pathway is critical to modulate energy expenditure in brown adipose tissue (Watanabe M. *et al. Nature* 2006). Thus, hepatic bile acid synthesis and bile acid pool size in the serum is critical to control metabolic rate. We clearly observed that several key genes involved in bile acid synthesis were largely downregulated in the HFD-fed $ROR\alpha^{LKO}$ mice. Also, serum total bile acid pool size was decreased. Though we still do not know the direct mechanism of how hepatic bile acid signaling was impaired in $ROR\alpha^{LKO}$ mice, at this stage we suggest that impaired bile acid signaling would be the key mediator in the liver-adipose axis to maintain systemic homeostasis. All these new data have been incorporated in revised supplementary Fig. 1.

4) In figure 4f and 4g, the first two bars are referring to the same experimental conditions. Therefore, why is the fold-increase in R.L.U. induced by $PPAR\gamma/PGC1\alpha$ so different in the two experiments (50 fold in panel f and less than 2 fold in panel g)?

The data have been replaced.

5) In figure 5c IgG or GFP (negative control Ab) condition is missing.

Negative control data have been incorporated.

6) Check primers list (some primers sequences are missing, e.g. Primers for *Cpt1b* mRNA expression).

The primer list has been provided as revised supplementary Table 5.

We thank the Reviewer for the very helpful comments, constructive suggestions, and requests for additional data, which have clearly enhanced the rigor of further supporting our hypothesis. We believe we have fully addressed them, and hope you like the revised manuscript.

Reviewer #3 (Remarks to the Author):

Kim et al developed and used hepatocyte-specific ROR α deficient mice, RNA seq, and ChIP-Seq to compile a narrative indicating a primary inhibitory role for ROR α in liver triglyceride storage and liver injury through an HDAC3-dependent mechanism that inhibits PPAR γ signaling.

The data are reasonably compelling, but like many manuscripts characterizing knockout models that have adiposity phenotypes, this study lacks differentiation of chicken from egg.

1. This is to say, does hepatocyte-specific deletion of ROR α increase caloric intake and/or decrease energy expenditure, causing obesity, insulin resistance, and fatty liver? Or are the observed phenotypes entirely attributable to effects of ROR α on PPAR γ signaling in a manner that regulates lipid metabolism in hepatocytes? Given the high fat diet induced obesity phenotype, the latter seems unlikely, and the authors do nothing to address this rather gaping hole.

We performed the metabolic cage to determine energy expenditure. We did not see any significant difference in control diet (CD)-fed ROR $\alpha^{fl/fl}$ and ROR α^{LKO} mice. However, we noticed that O₂ consumption and CO₂ production were largely decreased in HFD-fed ROR α^{LKO} mice. Thus, we suggest that environmental stress, such as HFD, leads to a decrease of systemic energy metabolism. We did not see any significant difference of food intake during the measurement. All updated data have been incorporated in revised Fig. 1 and supplementary Fig. 1.

How does hepatocyte-selective deletion of a transcription factor cause such significant reprogramming of adipose tissue and systemic energy homeostasis? This needs to be substantively addressed. The Discussion on sg mice is not particularly helpful, because as the authors recognize, this model lacks ROR α in all tissues including those of the nervous system.

Dr. Auwerx group has previously reported that bile acid signaling pathway is critical to modulate energy expenditure in brown adipose tissue (Watanabe M. *et al.*, *Nature* 2006). Thus, hepatic bile acid synthesis and bile acid pool size in the serum is critical to control metabolic rate. We clearly observed that several key genes involved in bile acid synthesis were largely downregulated in the HFD-fed ROR α^{LKO} mice. Also, serum total bile acid pool size was decreased. Though we still do not know the direct mechanism of how hepatic bile acid signaling was impaired in ROR α^{LKO} mice, at this stage we suggest that impaired bile acid signaling would be the key mediator in the liver-adipose axis to maintain systemic homeostasis. All these updated data and comments were incorporated in revised supplementary Fig. 1.

2. The experiment presented in Fig. 4e, Co-immunoprecipitation with ROR α and HDAC3, should also be performed examining endogenous proteins, not only those that are over-expressed. Clearly ROR α , HDAC3, and PPAR γ signaling are connected on several promoters, and the experiments presented in Fig. 4f and Fig. 5 help demonstrate this, but the data do not definitively support the competition model in Fig. 6h. Experiments to address would include HDAC3 knockdown and/or determination of HDAC3 recruitment in ROR α knockout liver.

We performed the experiments that the Reviewer requested. All data have been incorporated in revised Fig. 5a, Fig. 6c and supplementary Fig. 3d.

3. In Fig. 6, why does GW9662 decrease the floor on liver TAG and FAO gene regulation (6e-f), but not lipogenesis gene regulation (6g)? Specifically, there appears to be an ROR α independent component of the GW effect on the genes studied in 6f, and on liver TAG. This question addresses the greater concern that the authors may oversimplify the molecular mechanism among transcription factors studied.

We thank the Reviewer for the brilliant point. We have noticed that the expression of genes, including *Gck*, *Pepck*, *Srebp1c* and *Fasn* were decreased by GW9662 in WT mice from different mouse cohorts. To avoid oversimplifying our molecular mechanism, we determined other genes involved in lipogenesis/lipid sequestration, and we clearly showed that *Acc* and *Cidec* were largely decreased by GW9662 treatment compared with vehicle treated group. All data have been incorporated in revised Fig. 7g.

We thank the Reviewer for the very helpful comments, and requests for additional data, which have clearly enhanced the rigor of further supporting our hypothesis. We hope you like the revised manuscript.

In summary: we thank the Reviewers for their constructive suggestions and criticisms which have helped us make a more rigorous and readable manuscript. We were pleased that the response was favorable and believe that with the requested alterations, this revised manuscript is now fully suitable for *Nature Communications*.

REVIEWERS' COMMENTS:

Reviewer #2 (Remarks to the Author):

Kim et al. improved quite significantly the manuscript following the indications received from the reviewers. I still have a request to better address the issue of why treatment with TZDs leads to tangible clinical improvements of patients with fatty liver disease, whereas this study seems to suggest that hepatic PPAR γ activation is deleterious as it promotes lipid deposition in the liver. In the past, the "lipid steal" hypothesis has been proposed to explain why TZDs have beneficial effects on insulin resistance and T2D (Diabetes, 2003 vol.52 pp. 1311–1318; Diabetologia 2004 vol. Jul;47(7) pp.1306-1313). Could it be that the systemic effects of TZDs lead to improvement of hepatic steatosis because overall they favor lipid deposition in adipose tissue rather than ectopic deposition in other tissues like the liver and skeletal muscle? On the contrary, the local effects of hepatic PPAR γ activation promote ectopic deposition in the liver. In a few words, the effect of PPAR γ activation in adipose tissue offsets the effect of the local local activation of PPAR γ in the liver. In this context ROR α may play a role in modulating PPAR γ activity in the liver. This is a critical point of the manuscript and should be carefully and adequately discussed. I suggest to better address this point in the discussion to avoid misleading messages from this manuscript. The Methods section describing the experiment with the PPAR γ antagonist GW9662 does not indicate whether mice were on HFD, as it is stated in the legend to figure 7. I suggest to add this important experimental detail in the Methods section, specifying also for how long mice were fed with HFD and the vehicle used to add the antagonist to drinking water.

Reviewer #3 (Remarks to the Author):

The authors have satisfactorily responded to my concerns, and are to be commended.

Point-by-point response to the Reviewers' comments:

We thank the reviewers for their positive comments and for identifying some standing issues. Here we are submitting the final revised version of our manuscript based on their suggestions. Our detailed response to each of the reviewer's points is reported below.

To comply with #1 Reviewer's concerns:

We thank the Reviewer for highlighting this issue and for his/her suggestion. We have now addressed the possible contribution of PPAR α signaling network in the development of hepatic steatosis in evaluated the potential roles of ROR α on PPAR α signaling network.

Recent paper from Dr. David Moore's lab (Lee J.M. *et al. Nature* 2014) has shown that PPAR α is a nutrient sensing nuclear receptor to coordinate autophagic gene expressions. As we previously described in the manuscript, we tested the hepatic expression profiles of *Acox1* and *Fgf21* in ROR α^{ff} and ROR α^{LKO} mice and we did not find robust significant differences among genotypes in the setting of energy deprivation.

Hepatic steatosis is the pathological condition of liver to exhibit excessive lipid accumulation in the hepatocytes. And PPAR α is a well-known gene to control fatty acid beta oxidation. Thus, the PPAR α -null mice have been reported to develop severe hepatic steatosis. In our mouse model, PPAR α signaling has not been further increased in HFD-fed or fasted ROR α^{LKO} mice. However it is still possible that PPAR α signaling may be impaired or decreased to develop hepatic steatosis in HFD-fed ROR α^{LKO} mice. Thus, we proposed and modified our discussion to address that both abnormally upregulated PPAR γ transcriptional network and unknown PPAR α signaling that has been substantially impaired in HFD-fed ROR α^{LKO} mice may contribute to the development of hepatic steatosis in response to environmental stress such as high fat diet.

Nevertheless, we completely agree with the Reviewer's point to address the critical roles of PPAR α signaling in the development of hepatic steatosis. We believe that our description about the possible contribution of PPAR α signaling to develop hepatic steatosis would be suitable to explain the physiological roles of hepatic PPAR isoforms in the liver in response to high fat diet.

We thank the Reviewer for the very helpful comments, which have clearly enhanced the rigor of further supporting our hypothesis.

Reviewer #2 (Remarks to the Author):

Kim et al. improved quite significantly the manuscript following the indications received from the reviewers. I still have a request to better address the issue of why treatment with TZDs leads to tangible clinical improvements of patients with fatty liver disease, whereas this study seems to suggest that hepatic PPAR γ activation is deleterious as it promotes lipid deposition in the liver. In the past, the “lipid steal” hypothesis has been proposed to explain why TZDs have beneficial effects on insulin resistance and T2D (Diabetes, 2003 vol.52 pp. 1311–1318; Diabetologia 2004 vol. Jul;47(7) pp.1306-1313). Could it be that the systemic effects of TZDs lead to improvement of hepatic steatosis because overall they favor lipid deposition in adipose tissue rather than ectopic deposition in other tissues like the liver and skeletal muscle? On the contrary, the local effects of hepatic PPAR γ activation promote ectopic deposition in the liver. In a few words, the effect of PPAR γ activation in adipose tissue offsets the effect of the local local activation of PPAR γ in the liver. In this context ROR α may play a role in modulating PPAR γ activity in the liver. This is a critical point of the manuscript and should be carefully and adequately discussed. I suggest to better address this point in the discussion to avoid misleading messages from this manuscript.

The Methods section describing the experiment with the PPAR γ antagonist GW9662 does not indicate whether mice were on HFD, as it is stated in the legend to figure 7. I suggest to add this important experimental detail in the Methods section, specifying also for how long mice were fed with HFD and the vehicle used to add the antagonist to drinking water.

We thank the reviewer for his/her comments.

As the reviewer pointed out, the “lipid steal hypothesis” has been arisen to explain the mechanism of TZD to improve metabolic parameters. Systemic effects of TZDs have been accepted and clinically approved to improve insulin resistance and even fatty liver disease in human patients. As reviewer suggested, the systemic effects of TZD improve hepatic steatosis because lipid deposition occurs in adipose tissue rather than liver or other tissues. As TZD has been shown to increase adipogenesis in adipose tissue to ameliorate metabolic parameters, lipid steal hypothesis has been widely accepted in the field.

Unlike systemic activation by TZD, PPAR γ signaling in the liver has been debatable. Though TZD treatment improves hepatic steatosis in human patients, recent studies have reported that hepatic PPAR γ -deleted mice are prone to be resistant to hepatic steatosis. In this context, we here describe the fundamental role of ROR α to modulate or control the “hepatic” PPAR γ activation to protect against hepatic steatosis and obesity.

Quite interestingly, we noticed that GW9662 treatment clearly ameliorated metabolic parameters in HFD-fed ROR α ^{LKO} mice, suggesting that PPAR γ antagonism would be promising for the beneficial improvements in metabolic syndromes. Consistent with our GW9662 treatment, a recent study has also demonstrated that PPAR γ antagonism may

promote energy expenditure to protect against diet-induced obesity in animal model. The authors have shown that anticancer drug Gleevec serves as a PPAR γ antagonist to suppress of PPAR γ target gene expressions (Choi S.S. *et al.*, *Diabetes* 2016). Thus, it is still required to understand the molecular network of PPAR γ with other transcriptional factors to regulate whole body metabolism.

Altogether, our study clearly suggested that ROR α negatively regulates “hepatic” PPAR γ transcriptional network to maintain lipid homeostasis in response to environmental stress, such as high fat diet. We described the issue of PPAR γ activation commented from the Reviewer#2 in the discussion in detail.

For the method, we modified all texts complying with Reviewer’s suggestions.

Reviewer #3 (Remarks to the Author):

The authors have satisfactorily responded to my concerns, and are to be commended.

We really thank the Reviewer for the very helpful comments, constructive suggestions, and requests for additional data, which have clearly enhanced the rigor of further supporting our hypothesis.

In summary: we thank the Reviewers for their constructive suggestions and criticisms which have helped us make a more rigorous and readable manuscript. We were pleased that the response was favorable and believe that with the requested alterations, this revised manuscript is now fully suitable for *Nature Communications*.